# Over-Tokenized Transformer: Vocabulary is Generally Worth Scaling

**Hongzhi Huang** [1]  **Defa Zhu** [1]  **Banggu Wu** [1]  **Yutao Zeng** [1]  **Ya Wang** [1]  **Qiyang Min** [1]  **Xun Zhou** [1]

## Abstract

Tokenization is a fundamental component of large language models (LLMs), yet its influence on model scaling and performance is not fully explored. In this paper, we introduce Over-Tokenized Transformers, a novel framework that decouples input and output vocabularies to improve language modeling performance. Specifically, our approach scales up input vocabularies to leverage multi-gram tokens. Through extensive experiments, we uncover a log-linear relationship between input vocabulary size and training loss, demonstrating that larger input vocabularies consistently enhance model performance, regardless of model size. Using a large input vocabulary, we achieve performance comparable to double-sized baselines with no additional cost. Our findings highlight the importance of tokenization in scaling laws and provide practical insight for tokenizer design, paving the way for more efficient and powerful LLMs.

## 1. Introduction

The rapid advancements in large language models (LLMs) have been driven by innovations in model architectures (Vaswani et al., 2017) and training paradigms (Radford et al., 2019; Brown et al., 2020). Moreover, with the guidance of scaling laws (Kaplan et al., 2020), models prove to become stronger with increasing number of parameters or training data. Tokenization, the process of converting raw text into discrete tokens as the input and output of the model, is recently found relevant to scaling laws, where larger models deserve larger vocabulary and achieve better performance under the same training cost (Tao et al., 2024). In fact, expanding the input vocabulary incurs almost no additional computational cost, whereas expanding the output vocabulary significantly increases the training overhead for

[1]Bytedance Seed. Correspondence to: Hongzhi Huang <huanghongzhi.51@bytedance.com>.

*Proceedings of the 42nd International Conference on Machine Learning*, Vancouver, Canada. PMLR 267, 2025. Copyright 2025 by the author(s).

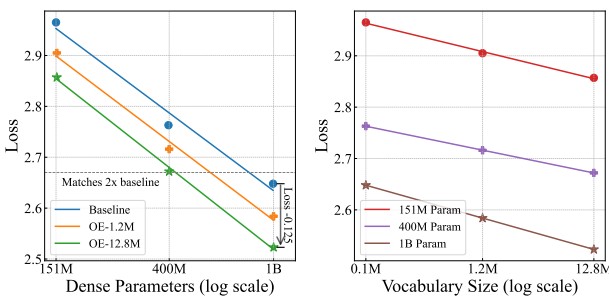

Figure 1: Scaling trend for Over-Encoded models and baselines on OLMo2. We plot the loss with 400B tokens' training. For over-encoding, input vocabulary size is extended from 0.1 to 1.2 and 12.8 million ($12\times$ and $128\times$ larger than baseline), referred to as OE-1.2M and OE-12.8M. We observe OE-12.8M with 400M parameters matches the baseline with 1B parameters.

smaller models. Thus, it is natural to consider decoupling the input and output vocabularies for separate investigations. Our research is fundamentally based on this idea.

Starting with synthetic experiments on context-free grammar modeling, we systematically analyze the effects of token granularity and vocabulary size on models of varying scales. Firstly, it is revealed that larger tokenizers improve the performance of larger models while introducing challenges for smaller ones, which is consistent with previous studies. Furthermore, once the input and output vocabulary is decoupled, we find that scaling up input vocabulary solely keeps improving model, while larger output vocabulary may be harmful to smaller models. This insight motivates the development of Over-Tokenized Transformers, which decouple the encoding and decoding vocabularies to achieve greater flexibility and performance gains.

Specifically, we introduce Over-Encoding(OE), which utilizes large hierarchical $n$-gram input vocabulary. As shown in Figure 1, our method improves the model scalability by a significant margin. Increasing the input vocabulary size by $128\times$, our 400M model matches the training loss of a 1B baseline with no additional training cost (see left panel). More interestingly, we observe a strong log-linear relationship between the input vocabulary size and model

performance, *i.e.*, exponentially increasing the input vocabulary size consistently results in a linear decrease in loss (see right panel). These findings represent a new dimension in scaling laws and also indicate embedding parameters as a new scalable sparse dimension. Moreover, we propose the concept of Over-Decoding(OD), which leverages larger output vocabulary to provide more fine-grained supervision. We typically treat multi-token prediction methods (Gloeckle et al., 2024; DeepSeek-AI et al., 2024) as approximations of OD. Combining OE and OD together, we build Over-Tokenized Transformer, which shows greater potential than apply either OE or OD solely.

Although introducing a large amount of embedding parameters for OE, the training and inference cost barely increases, as embedding parameters are used in an extremely sparse manner. In practice, we propose efficient engineering solutions to mitigate the computational and memory challenges introduced by large vocabularies, resulting an additional training overhead less than 5%.

Through this work, we aim to bridge the gap between tokenizer design and model scaling, positioning tokenization as a critical factor in the continued evolution of large language models.

## 2. Related Work

### 2.1. Tokenization Design

Tokenization is a critical component in the development of large language models (LLMs). Established methods such as Byte-Pair Encoding (BPE) (Sennrich, 2015) and Unigram Language Models (Kudo, 2018) have been widely adopted to create subword vocabularies that balance sequence length and vocabulary size. Recently, alternative paradigms have been proposed to address the computational inefficiencies of byte-level models (Xue et al., 2022; Clark et al., 2022; Tay et al., 2022; Yu et al., 2023). For instance, MegaByte (Yu et al., 2023) combines byte-level modeling with patching by first predicting patches and then predicting bytes within each patch, improving processing efficiency. However, its performance at scale remains inferior to that of tokenizer-based approaches. Building on this concept, In this work, our findings suggest that applying $n$-gram embeddings on top of BPE expands the vocabulary size, improving model capability, while BLT (Pagnoni et al., 2024) also has the same finding on byte-level models.

### 2.2. Scaling Vocabulary

Huang et al. (2021) firstly explores recurrent neural networks with scalable $n$-gram embedding table, where modular hashing is used to control vocabulary size. Roy et al. (2022) further augments the Transformer architecture with $n$-grams that are constructed from a discrete latent representation of the text sequence. Recent empirical studies have systematically investigated the relationship between vocabulary size and model performance. Tao et al. (2024) demonstrates that expanded vocabularies enhance both training efficiency and model performance, particularly in larger architectures. Building on these findings, we argue that vocabulary size research should separately consider embedding (input) and unembedding (output). While embedding incurs only lookup costs, unembedding introduces computational costs that scale with vocabulary size. More importantly, we find that input and output vocabularies exhibit distinct scaling behaviors, underscoring the need for independent scaling strategies to optimize model design. Moreover, Liu et al. (2024) revisits $n$-gram language modeling at massive scale, proposing an $\infty$-gram model trained on 5 trillion tokens, which achieves competitive performance and can effectively complement neural LMs to reduce perplexity.

### 2.3. Multi-Token Prediction

Multi-Token Prediction (MTP) (Stern et al., 2018; Qi et al., 2020; Gloeckle et al., 2024) has advanced the field of next-token prediction through the introduction of auxiliary objectives for simultaneous multi-token prediction. This methodology shares fundamental principles with our $n$-gram modeling framework, where multi-token prediction can be theoretically formulated as an approximation of $n$-gram unembedding utilization (over-decoded models). In our work, we further compare the effectiveness of MTP and over-encoded models, demonstrating that the performance gains from MTP and over-encoded models are complementary and can be combined to achieve even greater improvements.

## 3. Method

### 3.1. Insights from Synthetic Experiments

We initiate our research on tokenizers on a synthetic language modeling task to investigate their impact on model performance.

Following the experimental setup in the previous study (Allen-Zhu & Li, 2024), we employ a Context-Free Grammar (CFG) as the target language to generate sequences composed of 3 distinct characters, with sequence lengths up to 729. With this setup, the ground-truth language distribution is completely known, enabling precise evaluation of the language models. We train GPT-2 models (Radford et al., 2019) of various sizes on CFG-generated samples using next-token prediction loss and evaluate them based on the accuracy of model-generated sequences, which measures the proportion of valid generations. Additional details regarding the experimental setup can be found in Appendix B.2.

Our first experiment aims to compare the performance of

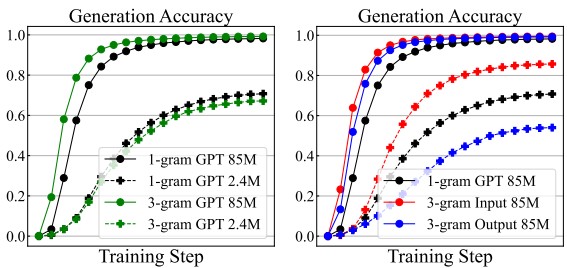

Figure 2: Performance comparison for models trained on CFG data. The left panel compares 1-gram and 3-gram tokenizers, showing that 3-gram improves larger (85M parameters) models but harms smaller (2.4M parameters) ones. The right panel examines 3-gram usage in encoders and decoders, revealing consistent gains with 3-gram encoders regardless of model size, while 3-gram decoders degrade performance in smaller models.

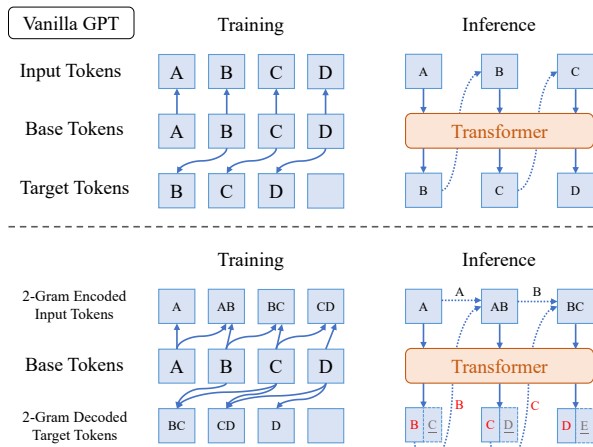

Figure 3: Illustration of 2-gram encoding/decoding GPT. Note that 2-gram decoding only preserves the predicted next 1 token though next 2 is predicted, which keeps inference cost identical to the vanilla model.

language models using tokenizers with varying levels of granularity. A baseline tokenizer constructs a vocabulary using the three terminal characters defined by the CFG, tokenizing sentences character-wisely, which we refer as a 1-gram tokenizer. We further define $n$-gram tokenizers, whose vocabulary comprises all $3^n$ possible combinations of $n$ sequential characters. We train both larger and smaller GPT-2 models using 1-gram and 3-gram tokenizers, respectively.

As illustrated in the left panel of Figure 2, we observe that a larger tokenizer improves the performance of larger models but negatively impacts smaller models. Notably, larger tokenizers result in shorter training sequences, which significantly reduces training cost. Thus, training larger models with a 3-gram tokenizer not only reduces training costs but also enhances model performance. An intuitive insight is that larger models benefit from larger vocabularies, improving both training efficiency and performance. This finding is also supported by previous studies (Tao et al., 2024).

To decouple the influences of enlarging input and output vocabularies, we separately introduce $n$-gram encoding and decoding models, as illustrated in Figure 3. To begin, the raw text is tokenized character-wisely via 1-gram tokenizer. In the $n$-gram encoding model, input tokens are converted to $n$-gram tokens at the input layer, making a large input vocabulary of size $3^n$, and the unembedding layer remains 1-gram, predicting next one character. In the $n$-gram decoding model, the input remains 1-gram tokens while the target labels (i.e., the next token) are converted to $n$-gram labels, resulting a fine-grained classification head that predicts the conditional joint distribution of next $n$ tokens. Note that the training sequence length remains unchanged and matches the length produced by the 1-gram tokenizer for both variants.

To maintain comparable inference costs, the $n$-gram output model does not generate $n$ tokens simultaneously. Instead, it samples an $n$-gram prediction but preserves the next 1 token only, disregarding extra token predictions during inference. The results for $n = 3$ of the two variants are shown in the right panel of Figure 2. We find that the two models exhibit different behaviors. The 3-gram encoding model consistently enhances performance across all model sizes. However, the 3-gram decoding model improves performance for larger models but degrades it for smaller ones.

We conclude that, when using large tokenizers, the large input vocabulary is always positive while the large output vocabulary can be negative for smaller models. We hypothesize that the difference lies in their respective roles: the input embedding is responsible for encoding the context into feature embeddings, where a larger vocabulary enhances the representational capacity of the feature mapping, thereby positively impacting the model. In contrast, the output vocabulary determines the granularity of the prediction task. A larger output vocabulary implies more fine-grained supervision signals, which can either be beneficial (e.g., for large models prone to overfitting) or burdensome (e.g., for smaller models suffering from severe underfitting). Motivated by this observation, we extend our exploration to over-tokenized transformers in real-world natural language modeling.

### 3.2. Over-Tokenized Transformer

Above all, we use a standard natural language tokenizer as the base tokenizer. Then, the challenge arises: if the base tokenizer has a vocabulary size of $V$, which is usually as large as $10^5$, the $n$-gram vocabulary with a size of $V^n$

becomes exceedingly large and impractical. To address this, we propose approximating the large embedding table through a series of matrix decompositions.

Given a sequence of input ids $x_1, x_2, \ldots, x_t$ from the base tokenizer, we define the $n$-gram input token $x_i^{(-n)}$ as follows:

$$x_i^{(-n)} = f(x_i, x_{i-1}, \ldots, x_{i-n+1}), \quad (1)$$

where $f(z_1, \ldots, z_n)$ is an index mapping function, and out-of-range indices are padded with zero tokens, *i.e.*, $x_i = 0, \forall i \notin [1, t]$. One intuitive design treats $(z_1, \ldots, z_n)$ as a $p$-base number, defining $f$ as

$$f(z_1, \ldots, z_n) = \sum_{i=1}^{n} z_i p^{i-1}, \quad (2)$$

where $p \geq V$ ensures that $f$ is a bijection. Typically, $p$ is set to $V$ to keep the range of $f$ as compact as possible. Notably, $x_i^{(-1)} = x_i$ corresponds to the standard transformer input.

**General $n$-gram Embedder.** The key to designing an flexible $n$-gram embedder module is to make the vocabulary size configurable. We achieve this efficiently using a simple tiled matrix parameterization approach. Specifically, the tiled matrix parameterization extends an $m \times d$ embedding table into a $V^n \times d$ embedding table by tiling, where $m$ is a configurable size. In practice, the lookup process is straightforward: an input token $x^{(n)}$ is mapped by taking its modulus with respect to $m$. In summary, our $n$-gram embedder is formalized as

$$\mathbf{h} = \mathbf{E}\left(x^{(n)} \% m\right), \quad (3)$$

where $\mathbf{h}$ is the output embedding, $\mathbf{E} \in \mathbb{R}^{m \times d}$ is the embedding matrix, and $\%$ is the modulo operation. We denote this $n$-gram $m \times d$ embedder as $\mathbb{E}^{m \times d}(x^{(n)})$. This $n$-gram hashing technique is commonly used in previous works (Huang et al., 2021; Roy et al., 2022).

**Over-Encoding(OE).** We find a hierarchical encoding paradigm to be highly effective. Specifically, we compute the input embedding to the GPT model as the sum of 1-, 2-, ..., $n$-gram embeddings. Additionally, we observe further benefits from using smaller embedding dimensions. An embedder $\mathbb{E}_n^{m \times d}$ can be sliced into $k$ low-rank decomposed embedders, represented as:

$$\mathbb{E}^{m \times d | k}(x^{(-n)}) = \sum_{i=1}^{k} \mathbb{E}_i^{m \times \frac{d}{k}}(x^{(-n)})\mathbf{W}_i, \quad (4)$$

where $\mathbf{W}_i \in \mathbb{R}^{\frac{d}{k} \times d_{model}}$ projects the embedding vector to match the model dimension. Using the same number of

embedding parameters and incurring only minimal additional cost through $k$ dense matrices $\mathbf{W}_i \in \mathbb{R}^{\frac{d}{k} \times d_{model}}$, this approach significantly enhances performance.

Overall, the over-encoding process maps an input token to an embedding as follows:

$$\mathrm{OE}(x) = \mathbb{E}^{V \times d}(x^{(-1)}) + \sum_{i=2}^{n} \mathbb{E}^{m \times \frac{d}{n} | k}(x^{(-i)}), \quad (5)$$

where the 1-gram embedding $\mathbb{E}^{V \times d}(x^{(-1)})$ is implemented consistently with the original Transformer to align with the tied weight design. Generally, $m$ is set to a value much larger than $V$, and the model performance is observed to improve consistently as $m$ increases.

Notably, for multiple embedders with $m$ rows, minor adjustments (*e.g.*, replacing $m$ with $m + 2$) are made to ensure each embedder has a unique mapping. This increases the combinatorial capacity of the embedding; otherwise, the slice trick would make no difference. A detailed pytorch-like implementation for OE can be found in Appendix A.

We also acknowledge a concurrent work, BLT (Pagnoni et al., 2024), which employs a similar $n$-gram hashing embedding strategy for byte-level tokens.

**Over-Decoding(OD).** Based on the conclusions from our CFG experiments, decoding additional tokens is only effective for sufficiently large models. As a matter of fact, previous researches on Multi-Token Prediction (MTP) (Gloeckle et al., 2024) are typically approximations of over-decoding, and share the same conclusion that only large models benefit from future token predictions. Generally, MTP-like methods are viewed as over-decoding in this paper. In addition, we explore other solutions for over-decoding in Appendix C for reference.

**Over-Tokenized Transformer(OT).** Integrating over-encoding and over-decoding, we obtain over-tokenized transformer. Specifically, we focus on the conditional recursive form of MTP proposed in DeepSeek V3 (DeepSeek-AI et al., 2024), which we refer as MTP-DS. In this formulation, MTP no longer predicts the next few tokens in parallel but instead predicts them sequentially. For the $n$-th head, the embedding of the next $(n - 1)$-th token is concatenated to the layer input as a condition for the next $n$-th token prediction.

Under MTP-DS architecture, over-encoding enhances the representation capacity of token embeddings and directly participates future token predictions. On the one hand, the future token prediction tasks become easier to learn. On the other hand, the over-encoding can be trained more sufficiently. With these advantages, the integration of the two methods yields greater benefits, even on relatively smaller models.

## 3.3. Engineering Challenges and Solutions

The over-encoding constructs a very large input vocabulary. In theory, as the embeddings are sparsely accessed based on token ids, enlarging vocabulary should barely impact the training or inference cost. However, the large embedding parameters can place substantial memory pressure on GPUs. Furthermore, when parameter sharding strategies, such as FSDP (Zhao et al., 2023), are applied during training, the communication of these sparse parameters can severely degrade training efficiency, further constraining the choice of $m$ (the vocabulary size) to smaller values.

To mitigate this issue, we recommend using tensor parallelism specifically for the over-encoding embedding layer to reduce communication overhead. The embedding table is row-wise sharded across all data-parallel (DP) ranks. For a given input, the token is sent to the corresponding DP rank that holds its embedding, queried for the embedding vector, and then the resulting embedding is sent back to the original DP rank. This process involves two all-to-all communications during the forward pass and one all-to-all communication during the backward pass, resulting in a total communication volume significantly lower than that of FSDP.

We implement this optimization and find that training an over-encoded model with $m = 10^7$ on FSDP reduces throughput by less than 5%. In contrast, without this optimization, FSDP experiences a 25% slowdown and easily runs out of memory for vocabulary sizes exceeding $m = 5 \times 10^6$.

We believe that the current implementation still does not represent the limit of over-encoding performance optimization. Its greatest advantage is that the vocabulary input is decoupled from the model architecture. This decoupling allows the embedding lookup for the next micro-batch to be performed in advance. For example, we can design a dedicated embedding lookup stage in a pipeline-parallel training framework, which overlaps the communication required for embedding lookups with the transformer forward of the current micro-batch. This strategy would maintain training throughput without any performance degradation. Additionally, under this approach, the over-encoding parameters could be offloaded to the CPU, completely alleviating GPU memory pressure. Notably, similar training frameworks have already been implemented (Fang et al., 2022; Li et al., 2023). Over-Encoding could leverage these designs to improve model performance with minimal additional cost.

## 4. Experiments

**Implementation.** Our experiments focus on large language modeling tasks and basically follow OLMo series works. We maintain the training configuration of baseline model, replacing the word embedding with over-encoding technique. The original tokenizer used in baseline models is preserved as the base tokenizer in our method. The experiments are run with engineering optimized implementation. We refer OE-$m$ to over-encoded models with $m \times d_{model}$ embedding parameters in total. Typically, we implement with $n = 3, m = 1.28 \times 10^7$, *i.e.* OE-12.8M, and variate $k$ according to $d_{model}$, making $\frac{d_{model}}{nk} \approx 256$.

**Metrics.** We report the average perplexities (PPL) and losses on the `c4_en-validation` dataset as the 'Eval PPL' or 'Eval Loss', along with the metrics for zero-shot evaluation on downstream benchmarks. We observe significant volatility in the zero-shot performance indicators for the datasets. For more reliable and consistent results, we mainly concern five stable tasks in our analysis: MMLU-Var, Hellaswag, ARC-Challenge, ARC-Easy and PIQA. Detailed descriptions on these tasks and more comprehensive evaluations can be found in Appendix B.

### 4.1. Over-Encoding Scaling Trend

**Dense Models** We follow the experimental setup described in OLMo2 (OLMo et al., 2024), which is currently the state-of-the-art fully open model. We maintain most of the training configurations, but modify the architecture to obtain models with 151M, 400M and 1B dense parameters, which is denoted by OLMo2-151M, OLMo2-400M and OLMo2-1B, respectively. We train OLMo2-151M and OLMo2-400M models with 400B tokens, and OLMo2-1B with 1T tokens. In addition to OE-12.8M models, we also run experiments for OE-1.2M to offer a rough scaling trend on vocabulary size.

The result is ploted in Figure 1. From comparisons across model scales, OE models hold consistent improvements from baseline models. Specifically, the OE-12.8M model achieves comparable performance to a baseline model that is 2.5 times larger in model scale (comparing the 400M OE model with the 1B baseline model). Moreover, the vocabulary size of OE-12.8M, OE-1.2M, and the baseline model (vocabulary size $V = 100278$) exhibits exponential growth, while their scaling curves maintain approximately equal spacing. This observation suggests a log-linear relationship between vocabulary size and model performance, which we further validate in the ablation studies.

We present the training dynamics for OLMo2-1B models in Figure 4. Compared to the baseline model, OE demonstrates consistently increasing performance gains, with an improvement of 0.12 at 400B tokens that expands to 0.14 at 1T tokens. From the perspective of convergence acceleration, training loss achieves a remarkable 5.7× speedup. Furthermore, in terms of downstream evaluation, OE exhibits substantial acceleration, achieving 3.2×, 3.0×, 2.6×,

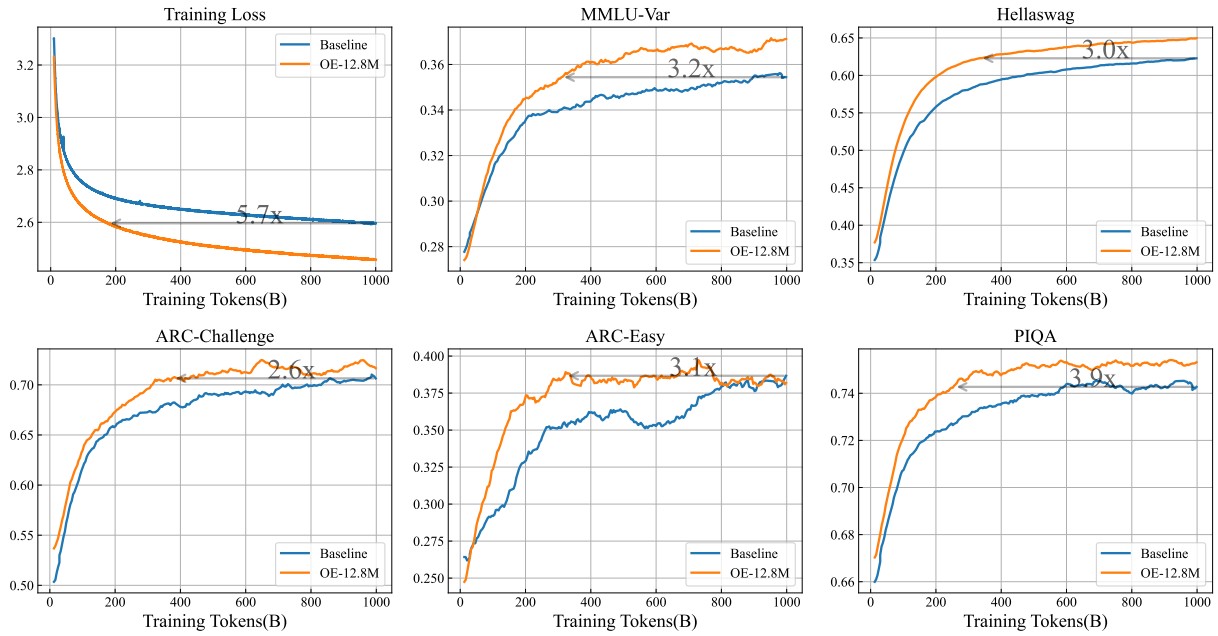

Figure 4: Training curves for OE-12.8M and baseline model on OLMo2-1B. The metrics are smoothed via exponential moving average with weight 0.99 for loss and 0.9 for downstream tasks. We observe significant convergence acceleration for the OE model: $5.7\times$ on loss, $3.2\times$ on MMLU-Var, $3.0\times$ on Hellaswag, $2.6\times$ on ARC-Challenge, $3.1\times$ on ARC-Easy and $3.9\times$ on PIQA.

Table 1: Performance of Over-Encoding on MoE architecture with 500B tokens' training. The column 'Emb. P.' represents 'Embedding Parameters'. 'Downstream' stands for the average of MMLU-Var, Hellaswag, ARC-Challenge, ARC-Easy, and PIQA. For '+OE' rows, we provide metric difference with blue labels.

| Model | # Emb. P. | Loss↓ | Downstream↑ |
|---|---|---|---|
| OLMoE-1.3B | 51M | 2.554 | 0.510 |
| +OE-12.8M | 13.1B | 2.472 -0.082 | 0.524 +0.014 |
| OLMoE-7B | 102M | 2.305 | 0.601 |
| +OE-12.8M | 26.3B | 2.229 -0.076 | 0.608 +0.007 |

$3.1\times$ and $3.9\times$ speedups on MMLU-Var, Hellaswag, ARC-Challenge, ARC-Easy and PIQA, respectively. A thorough evaluation is ploted in Appendix B.3.

**MoE Models** Sparse Mixture-of-Experts (MoE) models (Shazeer et al., 2017) have also achieved great success in recent years. These models aim to introduce sparse Feed-Forward Network (FFN) modules, enabling the addition of a large number of sparse parameters to improve model performance while keeping the same number of activated parameters, making the training and inference costs unchanged. Similarly, we evaluate the effectiveness of OE

under the MoE architecture.

We follow the experimental setup described by OL-MoE (Muennighoff et al., 2024). In the experiments, OLMoE-1.3B refers to a model with 260M active parameters and a total of 1.3B parameters, while OLMoE-7B refers to models with 1.3B active parameters and a total of 7B parameters. In this setting, the base tokenizer has vocabulary size $V = 50280$, and we train 500B tokens for all these models.

The results are shown in Table 1. We first compare the training loss. At two different model scales, OE-12.8M achieves approximately the same improvement in loss compared to the baseline, despite decreases in the proportion of embedding parameters as the model scales up (*i.e.*, $10\times$ dense parameters for OLMoE-1.3B and $3.7\times$ for OLMoE-7B). However, in terms of downstream evaluation metrics, the performance improvement of OE diminishes. We hypothesize that this reduction is related to the sparse parameters utilized in the MoE architecture, which may overlap with the benefits provided by sparse embedding parameters.

### 4.2. Ablation Study

We conduct ablation study basing on the setup of OLMoE-1.3B. To save computational resources, for ablation variants that are non-promising, we only train for 50B tokens. Other-

Table 2: Ablation study on different input vocabulary designs. The downstream tasks follow the eval settings in OLMoE, where MMLU-V stands for MMLU-Var, HS for Hellaswag, ARC-C for ARC-Challenge and ARC-E for ARC-Easy. All models are trained with 500B tokens.

| Id | Model | Train | | Eval | | Downstream | | | | |
|---|---|---|---|---|---|---|---|---|---|---|
| | | Loss↓ | PPL↓ | Loss↓ | PPL↓ | MMLU-V↑ | HS↑ | ARC-C↑ | ARC-E↑ | PIQA↑ |
| 1 | OLMoE-1.3B | 2.554 | 12.864 | 2.924 | 18.625 | 0.327 | 0.553 | 0.325 | 0.622 | 0.727 |
| 2 | $+\mathbb{E}^{3.2\text{M}\times d}(x^{(-2)})$ | 2.511 | 12.319 | 2.887 | 17.944 | 0.340 | 0.569 | **0.351** | **0.656** | 0.734 |
| 3 | $+\mathbb{E}^{6.4\text{M}\times\frac{d}{2}}(x^{(-2)})$ | 2.507 | 12.268 | 2.882 | 17.851 | 0.330 | 0.573 | 0.341 | 0.648 | 0.731 |
| 4 | $+\mathbb{E}^{3.2\text{M}\times d|2}(x^{(-2)})$ | 2.503 | 12.221 | 2.877 | 17.754 | 0.337 | 0.575 | 0.345 | 0.651 | **0.740** |
| 5 | $+\mathbb{E}^{3.2\text{M}\times d|4}(x^{(-2)})$ | 2.503 | 12.226 | 2.876 | 17.736 | 0.328 | 0.575 | 0.337 | 0.653 | 0.734 |
| 6 | $+\sum_{i\in\{2,3\}}\mathbb{E}_i^{3.2\text{M}\times\frac{d}{2}|2}(x^{(-i)})$ | **2.495** | **12.127** | **2.870** | **17.638** | **0.340** | **0.578** | 0.330 | 0.636 | 0.738 |
| 7 | $+\mathbb{E}^{12.8\text{M}\times d}(x^{(-2)})$ | 2.493 | 12.100 | 2.881 | 17.832 | 0.334 | 0.569 | **0.343** | 0.643 | **0.730** |
| 8 | $+\sum_{i\in\{2,3\}}\mathbb{E}_i^{12.8\text{M}\times\frac{d}{2}|2}(x^{(-i)})$ | **2.472** | **11.854** | **2.862** | **17.494** | **0.342** | **0.577** | 0.329 | **0.645** | 0.728 |

wise, we train for 500B tokens to obtain solid conclusions.

**Vocabulary Size Scaling.** We first analyze the scaling behavior of over-encoding based. Specifically, we use the simplest over-encoding setup with fixed $n = 2$ and $k = 1$, while varying the value of $m$. The size of $m$ ranges from 20K to 12.8M. We train on 500B tokens and evaluate the model performance. As shown in the figure, the experimental results reveal a logarithmic linear relationship between the training loss and $m$: for every $4\times$ increase in $m$, the training loss decreases by 0.015. Finally, we extracted the loss at 500B tokens and plotted the scaling curve shown in Figure 5. Here, the input vocabulary size is defined as the sum of the sizes of the 1-gram and 2-gram vocabularies, i.e., $V + m$. It is worth noting that the model needs to be sufficiently trained to capture these patterns. Figure 9 shows the training loss and loss diff curves in the vocabulary size scaling experiments. The larger vocabulary size requires longer training until the gains converge. And the performance gap is stably held after sufficient tokens' training.

**What's Good for OE.** We conducted extensive ablation studies on different configurations for the over-encoding. To ensure fairness, all configurations have the same number of embedding parameters. The results are shown in the Table 2. For simplification, we denote C-$i$ as configuration $i$ in the table.

We first validate that slicing the embedding table along the $d$ dimension yields further gains (comparing C-2, C-4, and C-5). Then, C-3 is constructed as a baseline, where we keep using one embedding table but scale up $m$ and scale down $d$. Such configuration also improves baseline but underperforms C-4 and C-5. We attribute this to the

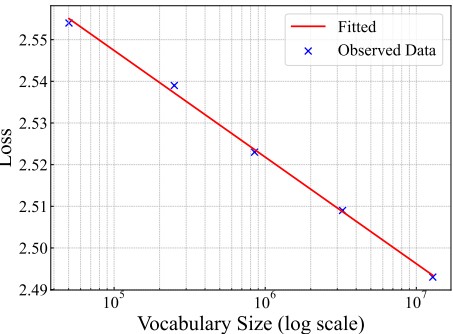

Figure 5: Log-linear relationship is observed between vocabulary size $m$ and training loss $\mathcal{L}$, *i.e.* $\mathcal{L} = 2.6754 - 0.0256 \times \log_{10} m$. The values are collected with 500B tokens' training on OLMoE-1.3B models.

reduced memory access: configuration 4 accesses $2d_{\text{model}}$ embedding parameters while configuration 3 accesses only $1.5d_{\text{model}}$ per token. Moreover, we validate the benefits of enlarging $n$ from 2 to 3 by comparing C-5 and C-6. With longer-range dependencies introduced, C-6 further improves performance. Lastly, these tricks are then applied on a larger vocabulary with $m = 12.8\text{M}$ (comparing C-7 and C-8). We observe even larger gains comparing to experiments with $m = 3.2\text{M}$. Note that C-6 and C-8 meets our default OE implementation, *i.e.*, OE-3.2M and OE-12.8M respectively, as described in (5).

**What's Bad for OE.** We also identify certain configurations that result in significant performance degradation and should avoid in practice. For this part of the experiments, due to the obvious performance drop, we report results af-

Table 3: Ablation study for the hierarchical design of over-encoding. The symbol '✓' on the $n$-gram column denotes $n$-gram token $x^{(-n)}$ is adopted. The experiments are conducted with $m = 3.2\text{M}$, and the metrics are reported after training on 50B tokens.

| 1-Gram | 2-Gram | 3-Gram | $k$ | Loss↓ | PPL↓ |
|:---:|:---:|:---:|:---:|:---:|:---:|
| ✓ | ✗ | ✗ | - | 2.714 | 15.094 |
| ✗ | ✓ | ✗ | 1 | 2.785 | 16.205 |
| ✓ | ✓ | ✗ | 1 | **2.678** | **14.555** |
| ✓ | ✓ | ✗ | 4 | 2.670 | 14.447 |
| ✓ | ✗ | ✓ | 4 | 2.684 | 14.642 |
| ✓ | ✓ | ✓ | 4 | **2.667** | **14.394** |

Table 4: Ablation study on hashing conflicts. Note the experiments are kept roughly the same vocabulary size, *i.e.* $64V \approx 3.218\text{M}$. The metrics are reported after training with 50B tokens.

| Model | Loss↓ | PPL↓ | Eval Loss↓ | Eval PPL↓ |
|:---|:---:|:---:|:---:|:---:|
| baseline | 2.714 | 15.094 | 3.094 | 22.060 |
| $+\mathbb{E}^{64V \times d}$ | 2.702 | 14.892 | 3.077 | 21.710 |
| $+\mathbb{E}^{3.2\text{M} \times d}$ | **2.678** | **14.555** | **3.054** | **21.202** |

Table 5: MTP Experiments on OLMoE-1.3B. The loss refers to the next one token prediction loss for MTP methods. Metric difference that improves baseline are marked blue while degrations are marked red.

| Model | Loss↓ | Eval Loss↓ | Downstream↑ |
|:---|:---|:---|:---|
| baseline | 2.554 | 2.924 | 0.510 |
| +MTP | 2.556 +0.002 | 2.925 +0.001 | 0.508 -0.002 |
| +MTP-DS | 2.555 +0.001 | 2.926 +0.002 | 0.511 +0.001 |
| OE-12.8M | 2.472 | 2.862 | 0.524 |
| OT-12.8M | 2.481 +0.009 | 2.869 +0.007 | 0.537 +0.013 |

### 4.3. Over-Tokenized Transformer

In this section, we verify the effectiveness of over-tokenized transformer. We conduct the experiments on OLMoE-1.3B with 500B tokens' training. As MTP is less efficient for smaller models, our implementation has only one extra head predicting the next 2 token, and set the weight 0.1 for future token loss. Then, the over-tokenized transformer is constructed, combining OE-12.8M with MTP-DS, which we refer as OT-12.8M.

Results are shown in Table 5. Limited by model capacity, MTP shows no improvement, as well as its stronger verison MTP-DS. However, with over-encoding plugged, MTP-DS presents greater power. OT further improves the downstream metric by 1. 3% compared to OE, although it slightly sacrifices the loss gains of OE. The detailed metrics are presented in Appendix B.4.

### 4.4. Speed Analysis

We analyze training and inference speed for over-encoding in this section. To illustrate training efficiency, we show training throughputs in OLMoE experiments in Table 6, where we run OE and baseline under the same hardware configurations. OE in OLMoE-7B yields more overhead as we did not carefully optimize engineering configurations.

| | OLMoE-1.3B | OLMoE-7B |
|:---|:---:|:---:|
| Hardware | 32×A100 | 64×A100 |
| Baseline | 1.211 | 0.494 |
| +OE 12.8M | 1.155 -4.63% | 0.453 -8.3% |

Table 6: Training throughputs for OE and baseline. We report average tokens per second in millions.

Theoretically, the additional flops introduced by OE is less than 0.5% (as shown in the table below), as demonstrated in Table 8. The overhead measured in the experiments should mainly be introduced by the all-to-all communication (which is proportional to the size of data parallel). And these

ter training on only 50B tokens, and simply use loss and perplexity as evaluation metrics.

First, we validate the importance of hierarchical multi-gram tokens. As shown in Table 3, removing the 1-gram vocabulary and using only the 2-gram vocabulary leads to a significant performance drop. This may be due to conflicts in the lookup results of the 2-gram table when 1-gram information is absent, making it difficult for the model to disambiguate the current semantics. On the other hand, skipping the 2-gram tokens and using only 1-gram and 3-gram tokens still shows gains over the baseline but underperforms the proposed model using 1-, 2-, and 3-gram tokens together. Overall, we hypothesize that the model benefits from hierarchical encoding to handle conflicting entries in the embedding table, thereby accurately resolving semantics.

Another supporting observation is that when we deliberately introduce more encoding conflicts (by setting $m$ to be an exact multiple of $V$), the gains of over-encoding are significantly reduced, as shown in Table 4. This observation also suggests that $m$ should set to a value coprime with $V$ to avoid such degeneration.

| Batch Size | Phase | Dense-1B | | MoE-1B/7B | | Dense-7B | |
|---|---|---|---|---|---|---|---|
| | | Baseline | OE-12.8M | Baseline | OE-12.8M | Baseline | OE-12.8M |
| 1 | Prefill | 20728.7 | 19446.4 | 6303.8 | 6189.0 | 6571.0 | 6499.9 |
| | Decode | 136.2 | 126.6 | 28.2 | 27.9 | 65.1 | 63.3 |
| 8 | Prefill | 36907.5 | 35902.6 | 22297.3 | 22292.1 | - | - |
| | Decode | 797.2 | 760.9 | 184.9 | 181.0 | 232.1 | 228.6 |
| 64 | Decode | 1422.1 | 1407.4 | 860.3 | 826.7 | - | - |

Table 7: Inference speed for OE and baseline. The sequence length is fixed to 2048, and we report tokens per second for prefilling and decoding separately under various batchsizes. Settings that cause OOM are leave blank.

| | OLMoE-1.3B | OLMoE-7B |
|---|---|---|
| baseline | 0.5409 | 2.3578 |
| +OE 12.8M | 0.5430 +0.38% | 2.3662 +0.35% |

Table 8: GFLOPs per token in the forward pass.

communication overheads can be further optimized through engineering techniques in the future.

For inference efficiency, we test the prefill and decoding throughput on a single A100 GPU using the `transformers` library. For the OE models, the additional embedding parameters are offloaded to the CPU, **incurring no GPU memory overhead**. The numeric results are shown in Table 7. The impact of OE on inference throughput is negligible, especially for larger models or larger batch sizes. In contrast, the sparse parameters introduced by MoE face severe memory access bottlenecks during inference. A very large batch size is required for the MoE model to achieve the same throughput as a dense model with the same activated parameters. Considering that the model inference might be carried out on more cost-effective but less computationally powerful inference GPUs, the relative overhead of OE could be further reduced.

## 5. Conclusion

In this work, we have explored over-tokenized transformer for language modeling. By systematically analyzing the impact of token granularity and vocabulary size across models of varying scales, we uncover an important scaling phenomena that inspires tokenizer design. Our findings reveal that larger input vocabularies consistently enhance model performance, no matter model scales, while larger output vocabulary can be harmful and difficult to learn for smaller models. These insights have significant implications for the continued scaling of LLMs. With such inspirations, we introduce over-encoding technique that scales up the input vocabulary via multi-gram token embeddings. Moreover, we develop over-tokenized transformer combining over-encoding and multi-token prediction. Extensive experiments validated

the effectiveness of our methods, showcasing significant gains in model performance. Notably, we demonstrated a strong log-linear relationship between input vocabulary size and training loss, highlighting a new dimension in the scaling laws of LLMs. These results emphasize the importance of tokenizer design as a key driver of advancements in language modeling.

In summary, this work bridges the gap between tokenizer design and model scaling, positioning tokenization as a critical factor in the development of next-generation language models. We believe that the proposed Over-Tokenized Transformers framework and the insights derived from our study will inspire further research into tokenization strategies and their role in scaling LLMs efficiently and effectively.

## Acknowledgements

This research was conducted at ByteDance Inc. We are grateful for the suggestions and discussions from Jundong Zhou, Zihao Huang, Yu Bao, Yike Yuan, Sijun Zhang and other colleagues at ByteDance Seed.

## Impact Statement

This paper presents work whose goal is to advance the field of Machine Learning. There are many potential societal consequences of our work, none which we feel must be specifically highlighted here.

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

# A. Pytorch Implementation

We provide a pytorch-like pseudocode for Over-Encoding in Algorithm 1.

---

**Algorithm 1** Over-Encoding in a PyTorch-like style.

---

```
#OE parameters:
# m: OE vocabulary size
# k: split num
# n: the number of neighboring tokens involved
#Model parameters:
# V: base vocabulary size
# D: model dimension

#Torch Modules
wte = nn.Embedding(V, d)
oe_embedders = nn.ModuleList([nn.Embedding(m+i*2, D//(k*(n-1)) for i in range(k*(n-1))])
oe_projs = nn.ModuleList([nn.Linear(D//(k*(n-1)), D) for _ in range(k*(n-1))])

def forward(self, input_ids):
    # input_ids: [bs, seqlens]
    x = self.wte(input_ids)
    n_gram_ids = input_ids.clone()
    for i in range(2, n+1):
        n_gram_ids += F.pad(input_ids, i-1) * V ** (i-1)
        for j in range(k):
            index = (i-2)*k+j
            x_oe = oe_embedders[index](n_gram_ids % (m + 2 * index))
            x += oe_projs[index](x_oe)
    x /= 1 + k * (n-1)
    return x
```

---

# B. More Experimental Details

## B.1. Downstream Benchmarks

We give a detailed description for the major benchmarks used in our experiments.

piqa (Bisk et al., 2020): a benchmark designed to evaluate models on commonsense physical reasoning. It tests the ability of models to reason about everyday physical interactions and affordances, requiring both textual understanding and physical intuition to solve tasks effectively.

hellaswag (Zellers et al., 2019): a dataset that focuses on testing commonsense reasoning and the ability to predict the most plausible continuation of a given scenario. It is considered a challenging benchmark due to adversarially filtered incorrect options aimed at confusing models.

arc_easy (Clark et al., 2018): a subset of the AI2 Reasoning Challenge (ARC) benchmark that contains grade-school-level multiple-choice questions. The questions in this subset are relatively straightforward and assess basic scientific knowledge and reasoning.

arc_challenge (Clark et al., 2018): the more difficult subset of the ARC benchmark, comprising scientifically challenging multiple-choice questions that require advanced reasoning, background knowledge, and problem-solving skills beyond basic memorization.

mmlu (Hendrycks et al., 2021) (Massive Multitask Language Understanding): a large-scale benchmark designed to test models across a wide range of academic and professional fields. It includes multiple-choice questions from diverse domains, such as humanities, STEM, and social sciences, aiming to measure comprehensive language understanding and reasoning capabilities.

## B.2. CFG Experiments

**Detailed Setup.** We obtain the grammar following (Allen-Zhu & Li, 2024) with the configuration named cfg3f, as illustrated in Figure 6. 20 million sentences are sampled from the grammar, serving as a fixed training dataset. We use standard GPT2 architecture, where the larger model uses a hidden size of 768 and the smaller model 128. Both larger and smaller models have 12 transformer layers, as depth is discovered essential to complex CFG modeling (Allen-Zhu & Li, 2024). We use AdamW optimizer with $\beta = (0.9, 0.98)$, weight decay 0.1, initial learning rate $3e^{-4}$ and batch size $64 \times 8$. The models are trained with cosine learning rate scheduler for 10 epoch. To evaluate model performance, we 10,000 sample

| root \|->20 21 | 19\|->18 16 18 | 16\|->15 15 | 13\|->11 12 | 10\|->8 9 9 | 7\|->2 2 1 | *an example sentence* |
|---|---|---|---|---|---|---|
| root \|->20 19 21 | 19\|->17 18 | 16\|->13 15 13 | 13\|->12 11 12 | 10\|->9 7 9 | 7\|->3 2 2 | |
| root \|->21 19 19 | 19\|->18 18 | 16\|->14 13 | 13\|->10 12 11 | 10\|->7 9 9 | 7\|->3 1 2 | 33221312331211312321132231231211121321132231131 |
| root \|->20 20 | 20\|->16 16 | 16\|->14 14 | 14\|->10 12 | 11\|->8 8 | 7\|->3 2 | 32233312312111213113311213212133333123221213123 |
| | 20\|->16 17 | 17\|->15 14 13 | 14\|->12 10 12 | 11\|->9 7 | 8\|->3 1 1 | 221111213322131131131131111113231233133133311331 |
| | 20\|->17 16 18 | 17\|->14 15 | 14\|->12 11 | 11\|->9 7 7 | 8\|->1 2 | 33333223121131112122111121123331233112111331333 |
| | 21\|->18 17 | 17\|->15 14 | 15\|->10 11 11 | 12\|->7 9 7 | 8\|->3 3 1 | 3311233331311113333121132113121211333332121111212 |
| | 21\|->17 16 | 18\|->14 15 13 | 15\|->11 11 10 | 12\|->9 8 | 9\|->1 2 1 | 21322322332213322111322113232331311121322322322 |
| | 21\|->16 17 18 | 18\|->15 13 13 | 15\|->10 10 | 12\|->8 8 9 | 9\|->3 3 | 21113333112132222133221121213312133133221122132 21 |
| | 21\|->16 18 | 18\|->13 15 | 15\|->12 12 11 | | 9\|->1 1 | 211213331232233312 |

Figure 6: **Left Panel**: CFG rules used in our experiments; **Right Panel:** an example of the generated sequences using the rules. This figure is taken from (Allen-Zhu & Li, 2024).

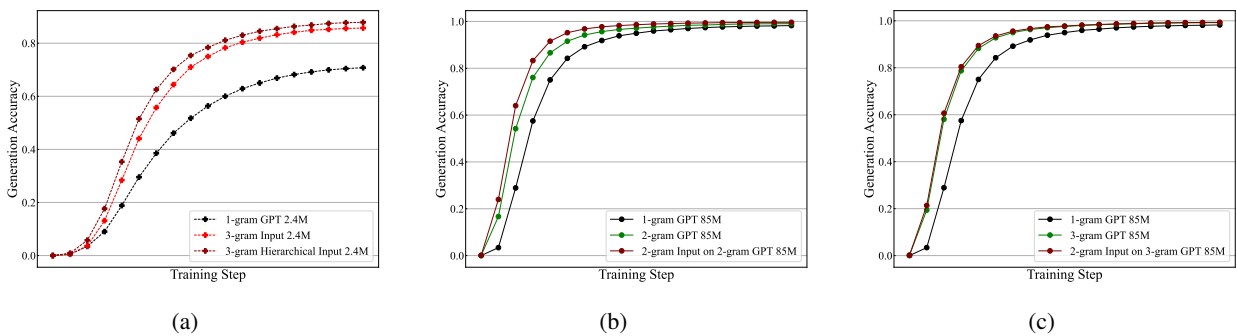

(a)  (b)  (c)

Figure 7: Performance comparison for models trained on CFG data.

sentences from the trained model auto-regressively using the raw next token probability. The sentences are then verified by the ground-truth grammar, and the ratio of accurate samples is denoted as generation accuracy.

**Implementation of $n$-gram tokenization or over-tokenization.** Though we mentioned a $3^n$-sized vocabulary for $n$-gram tokenization or $n$-gram over-tokenization in the main text, the actual vocabulary should be $2 + \sum_{i=1}^{n} 3^i$. The bias 2 stands for the BOS(Begin Of Sequence) and EOS(End of Sequence) token and extra $3^i$ terms are for the corner cases where the sequence length may not be divisible by $n$. We refer to $3^n$-sized vocabulary in the main text primarily to emphasize the order of magnitude.

**Hierarchical $n$-gram input further improves.** We experiment with introducing hierarchical $n$-gram inputs in the CFG setting. Initially, we use a single $V^n$-size $n$-gram embedding table to handle the model's input. To consider hierarchical $n$-gram modeling, we add $n - 1$ additional embedding tables, where the $i$-th table has a size of $V^i$, and use $i$-gram token as input. Notably, the parameters of these $n - 1$ embedding tables can be fully merged into the $n$-gram embedding table, so the effective parameter count does not increase. Instead, this process re-parameterizes the embedding table to decompose parameters hierarchically. The experimental results, shown in Figure 7(a), indicate that the model with hierarchically decomposed 3-gram inputs outperforms the model with naive 3-gram inputs. This further validates that even for models with a complete $n$-gram vocabulary (rather than approximated through matrix decomposition), hierarchically modeling multi-granularity tokens remains highly effective.

**Ablation on base tokenizers.** We replaced the base tokenizer to investigate whether further increasing the input vocabulary size could lead to additional improvements. Specifically, we expanded the input vocabulary for 2-gram and 3-gram base tokenizers to 2-gram inputs with vocabulary sizes of $(3^2)^2 = 81$ and $(3^3)^2 = 729$, respectively. The experimental results, shown in Figure 7(b) and Figure 7(c), demonstrate that increasing the input vocabulary consistently improves model performance. Another observation is that, when using the 3-gram base tokenizer, the benefit of increased convergence speed from 2-gram inputs becomes notably smaller. We hypothesize that this is related to the "slow-start" nature of large vocabularies. Similar observations have been made in natural language processing, where larger input vocabularies require more training tokens to realize their expected benefits.

### B.3. OLMo2 Experiments

**Detailed metrics.** We present a detailed training dynamics comparison for models on OLMo2-1B in Figure 11. OE achieves significant improvements across most metrics, while maintaining at least parity on the remaining ones.

### B.4. OLMoE Experiments

**Loss curves for vocabulary scaling.** We present the loss curves and loss difference curves in Figure 9. The larger input vocabulary requires longer training to obtain all its gains.

**Loss curves for OLMoE-7B models.** We present the loss curves for OLMoE-7B in Figure 10. The gains of OE have not fully converged in 500B tokens' training. Comparing to the experiments on OLMoE-1.3B, it is likely that larger models require more training to fully leverage the power of over-encoding.

**Detailed metrics for OE.** We present a detailed training dynamics comparison for OE-12.8M and baseline model in Figure 11(OLMoE-1.3B) and Figure 12(OLMoE-7B). Our model has significant improvements on most evaluation metrics, and has at least comparable performance for the rest.

**Detailed metrics for OT.** We present detailed metrics comparing OT-12.8M and OE-12.8M on OLMoE-1.3B in Figure 13. OT is not always better than OE, *e.g.*, Winogrande and Social IQA.

### B.5. In-house Experiments

We also perform in-house experiments to verify the effectiveness of over-encoding. The in-house baseline adopts MoE architecture, having 400M activated parameters and 4B sparse parameters in total. Then we implement OE, scaling up the vocabulary size to $m = 36M$. We train 600B tokens with in-house pre-training datasets, and evaluate the model through a series of open benchmarks under the few-shot settings. Results are shown in Table 9. OE shows significant gains across different domains, including reasoning, knowledge and math ability. Especially, we notice that OE mostly improves knowledge ability. These results demonstrate that OE also improves in-context learning.

Table 9: Comparison of OE and baseline on in-house models. The models are evaluated on various benchmarks on the few-shot settings. OE improves both reasoning, knowledge and math related benchmarks.

| | Reasoning Related Benchmarks | | | | |
|---|---|---|---|---|---|
| | ARC Challenge | Drop | BBH | WinoGrande | Hellaswag |
| Baseline | 65.7 | 34.4 | 37.1 | 63.2 | 66.2 |
| +OE 36M | 67.9 | 36.3 | 39.5 | 65.5 | 67.2 |

| | Knowledge Related Benchmarks | | | | |
|---|---|---|---|---|---|
| | MMLU | C-Eval | TriviaQA | MMLU-Pro | AGIEval |
| Baseline | 54.8 | 61.3 | 39.7 | 21.1 | 39.1 |
| +OE 36M | 57.9 | 68.3 | 49.0 | 24.1 | 43.2 |

| | Math Related Benchmarks | | |
|---|---|---|---|
| | Ape210K | GSM8K | MATH |
| Baseline | 63.7 | 40.6 | 22.7 |
| +OE 36M | 63.8 | 46.2 | 25.3 |

## C. Over-Decoding

**Method.** From previous experiments on synthetic data, we know that an excessively large output vocabulary is detrimental to small models. Therefore, when designing an over-decoding method based on natural language, we aim to leverage the advantages of $n$-gram prediction while keeping the decoding vocabulary size as small as possible. We found that product decomposition provides a promising solution to this problem.

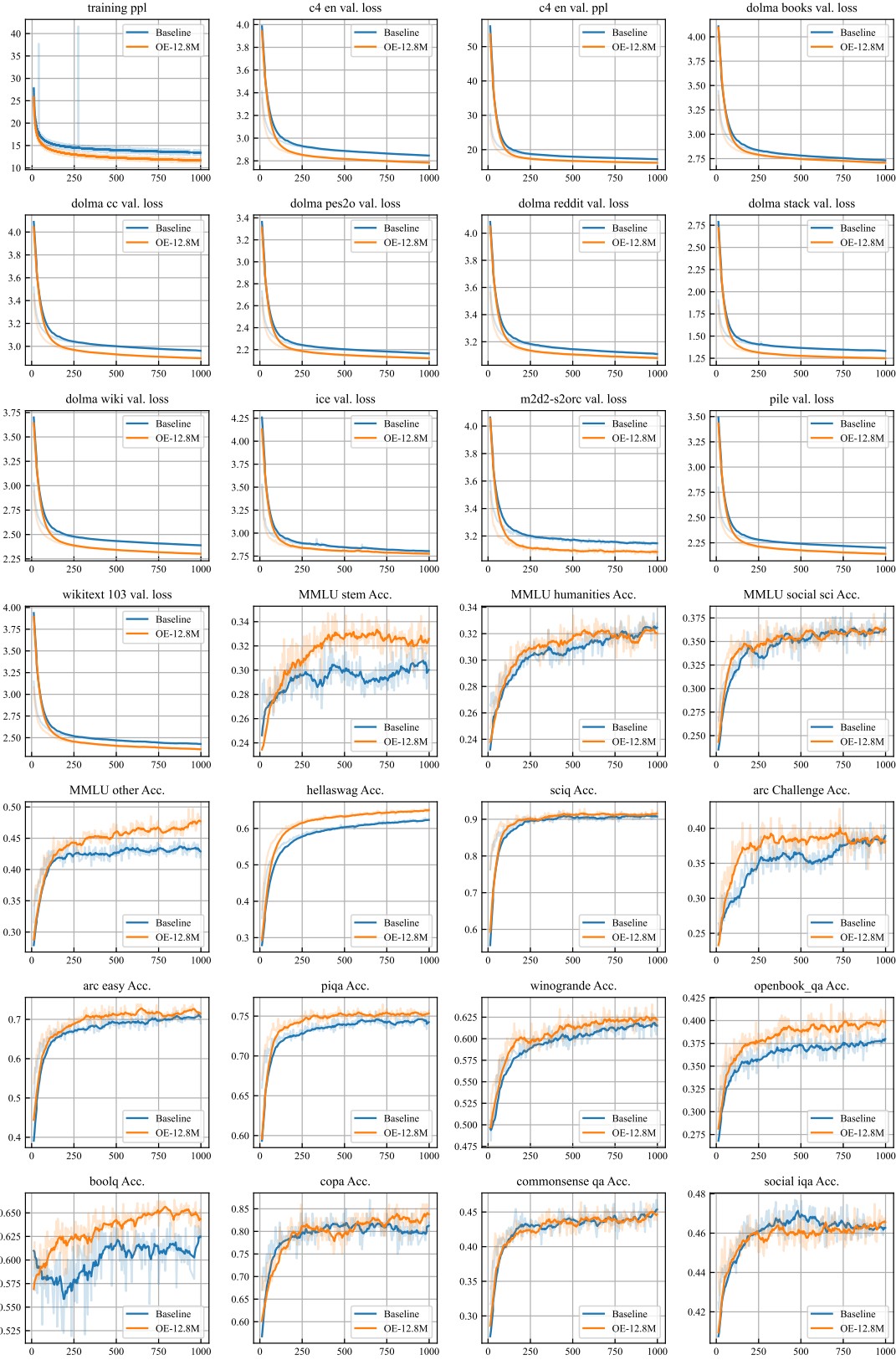

Figure 8: All metrics for OLMo2-1B, comparing OE-12.8M and baseline.

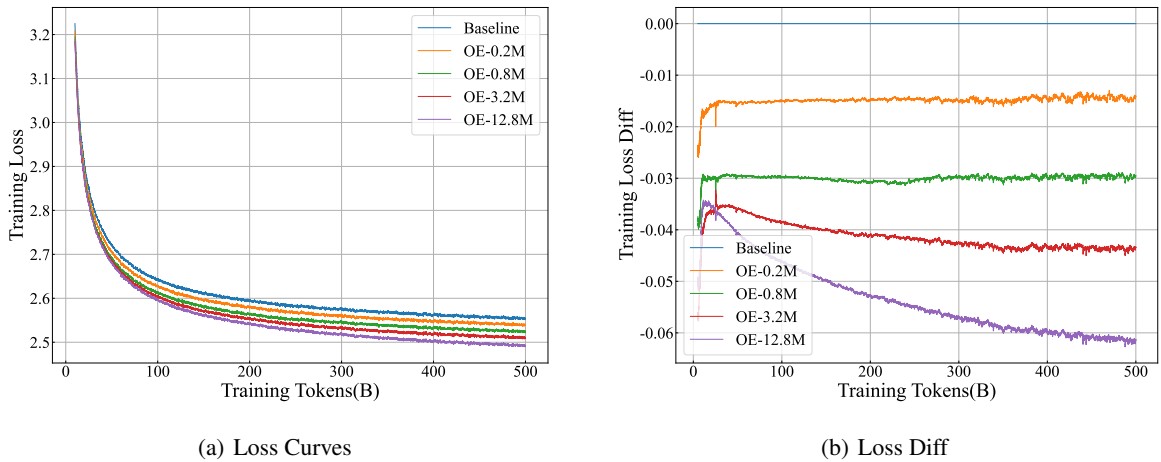

(a) Loss Curves

(b) Loss Diff

Figure 9: Loss curves for vocabulary scaling experiments, where OE models use the setting of $n = 2, k = 1$. The curves are smoothed with exponential moving average of weight $0.99$.

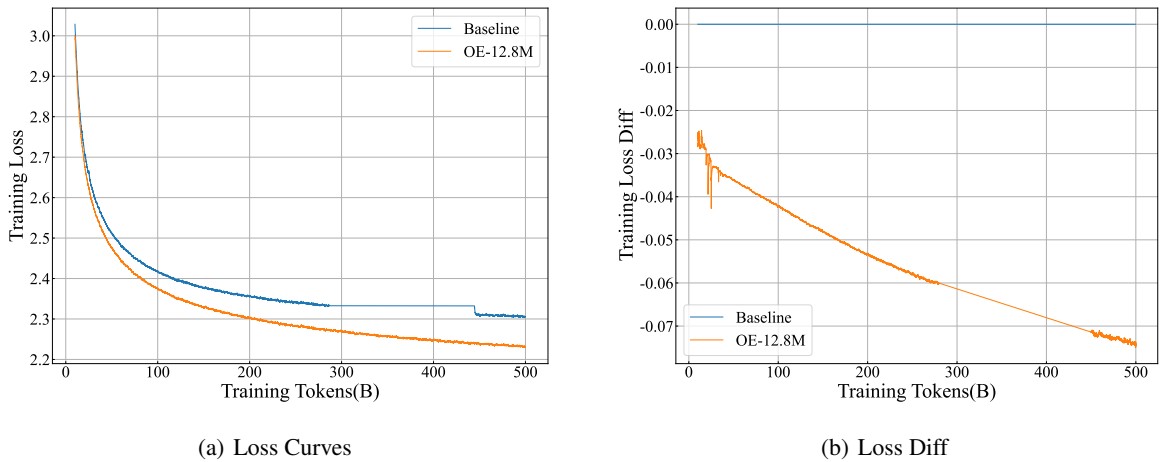

(a) Loss Curves

(b) Loss Diff

Figure 10: Loss curves for OLMoE-7B experiments. The OE model uses the setting of $n = 3, k = 4$. The curves are smoothed with exponential moving average of weight $0.99$. The flat part in baseline is a consequence of wandb log missing.

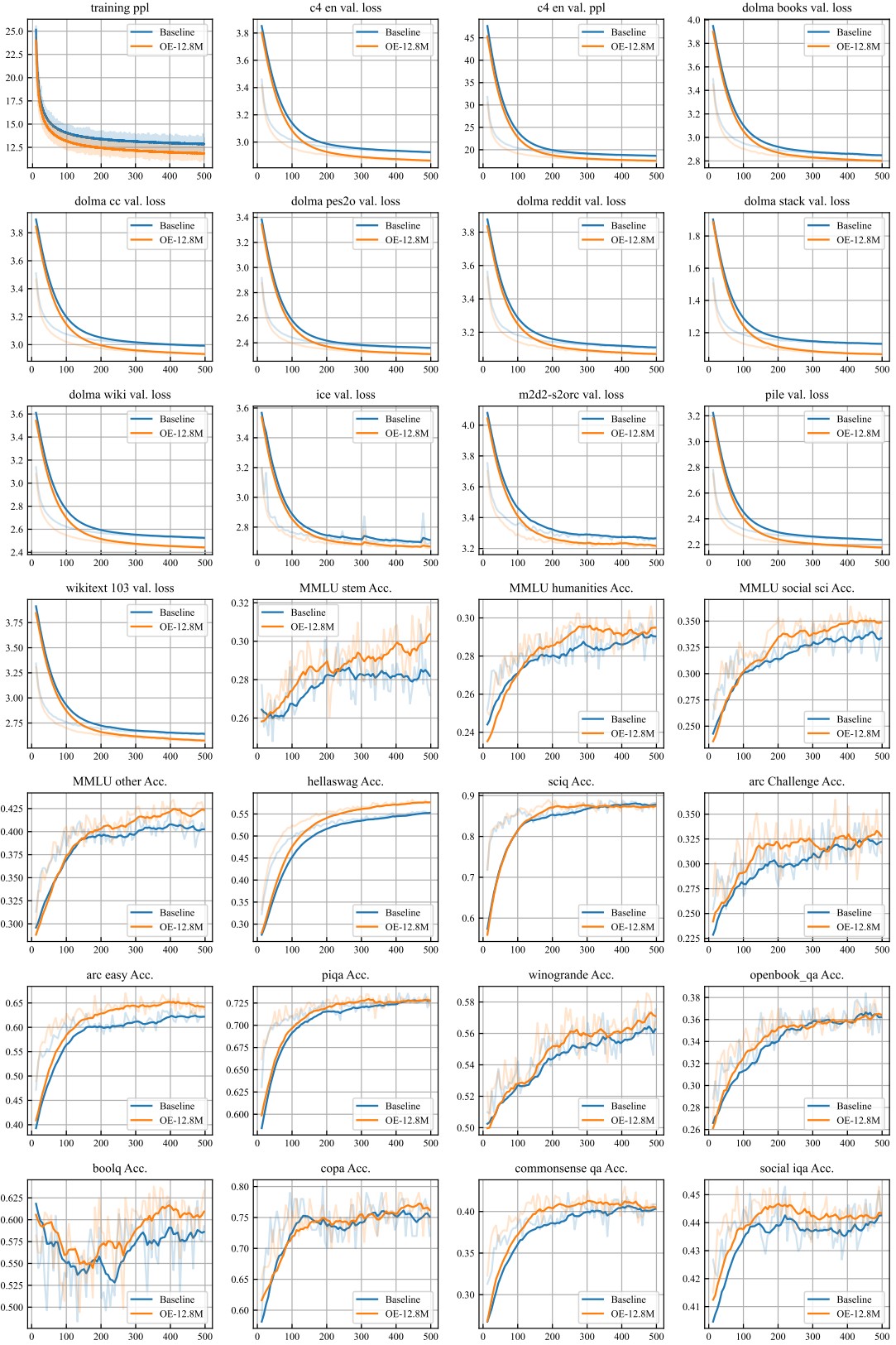

Figure 11: All metrics for OLMoE-1.3B, comparing OE-12.8M and baseline.

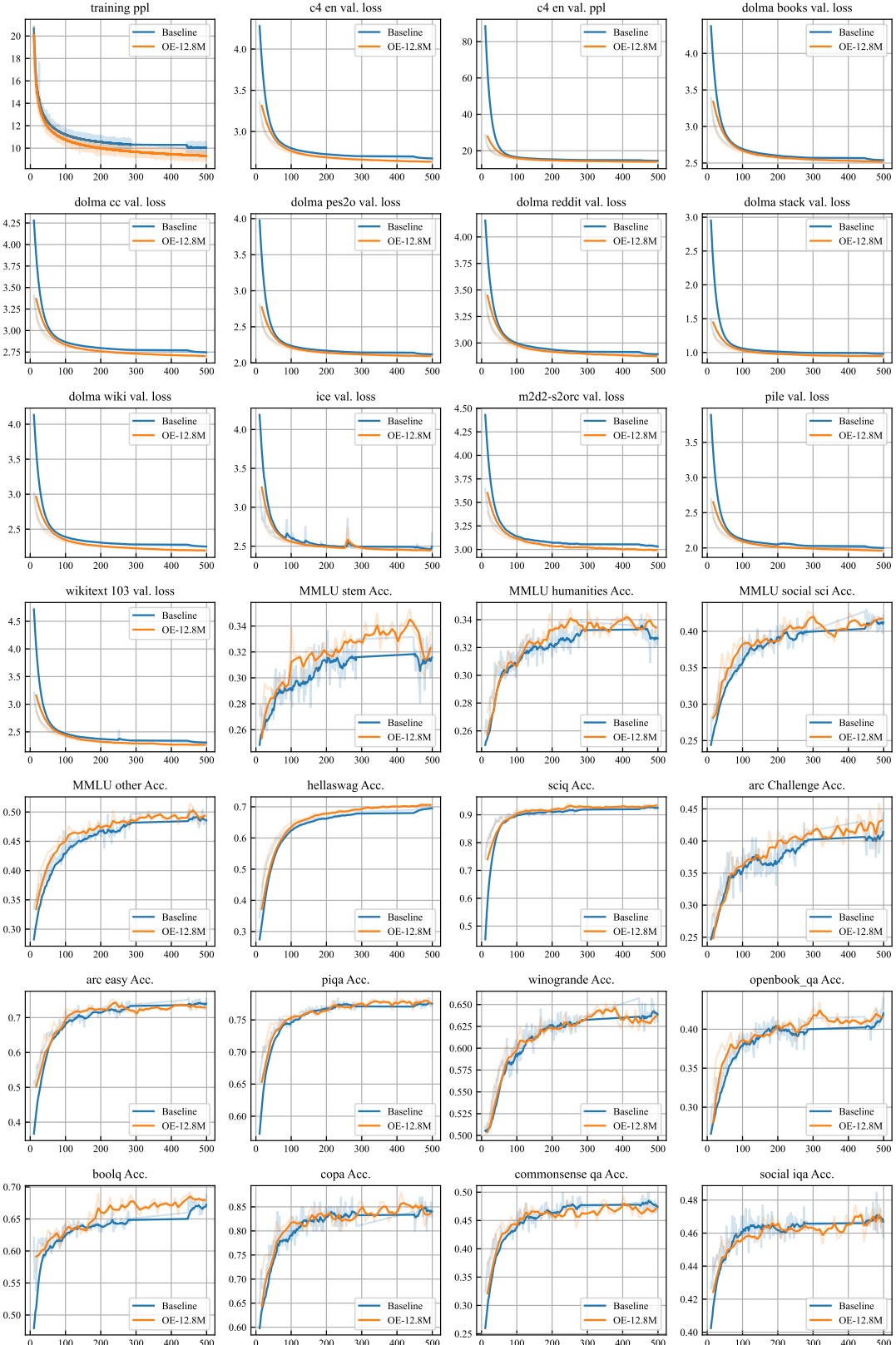

Figure 12: All metrics for OLMoE-7B, comparing OE-12.8M and baseline.

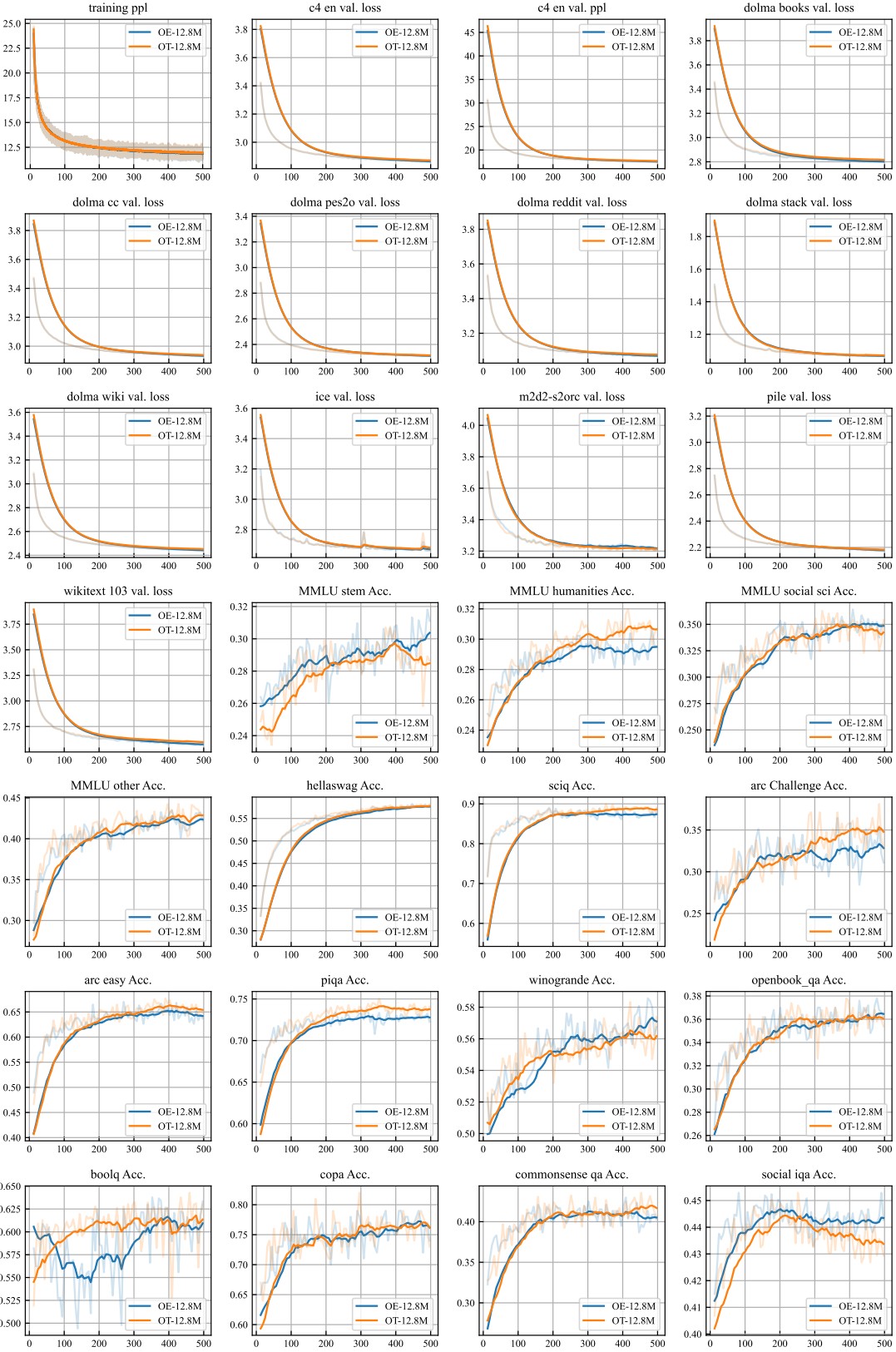

Figure 13: All metrics for OLMoE-1.3B, comparing OT-12.8 and OE-12.8M.

For the $n$-gram decoding vocabulary, the product embedding decomposition parameterizes a $V^n$-sized embedding table using $n$ separate $V$-sized embedding tables, formulated as:

$$\tilde{\mathbf{e}}_x = \sum_{j=1}^{n} \mathbb{E}_j^{V \times d}(x/V^{j-1}) = \sum_{j=1}^{n} \mathbf{E}_j(z_j), \tag{6}$$

where $\tilde{\mathbf{e}}_x$ is the reparameterized 2-gram embedding, and $z_1, \ldots, z_n$ correspond to the reverse mapping of the function $f$ given $x$, as described in 2. Then, during training, we compute the next $n$-gram token prediction loss on the reparameterized embeddings $\{\tilde{\mathbf{e}}_i\}$:

$$\mathcal{L} = \text{CE}(\hat{\mathbf{h}}\tilde{\mathbf{E}}^\top, x^{(+n)}) = \sum_{j=1}^{n} \text{CE}(\hat{\mathbf{h}}\mathbf{E}_j^\top, z_j), \tag{7}$$

where $\hat{h}$ represents the output hidden state, $\text{CE}(\cdot, \cdot)$ denotes the cross-entropy loss, and the loss decomposition is detailed later.

A notable challenge is that the decoding embedding is densely activated, making the unembedding layer computationally expensive, particularly for smaller models. To address this, we optionally apply a low-rank decomposition by projecting the last hidden state to a smaller dimension. Additionally, we reweigh these losses using a set of hyperparameters $\lambda_1, \ldots, \lambda_n$, resulting in the final training loss:

$$\mathcal{L} = \sum_{j=1}^{n} \lambda_j \text{CE}(\hat{\mathbf{h}}\mathbf{W}_j\mathbf{E}_j^\top, z_j), \tag{8}$$

where $\mathbf{W}_j$ is the optional projection matrix corresponding to the $j$-th embedding table. We typically set $\lambda_1 = 1$ and $\forall i > 1, \lambda_i \leq 1$. A pytorch-like implementation is provided in Algorithm 2.

**Derivation of Loss Decomposition** We provide a detailed derivation of the loss decomposition in (7). Recall that $\{\tilde{\mathbf{e}}_i\}$ are the $n$-gram token embedding table. Each of them is reparameterized by $\tilde{\mathbf{e}}_i = \sum_{j=1}^{n} \mathbf{E}_j(z_j)$ with $i = \sum_j z_j V^j$. We denote $\mathbf{e}_{z_j}^{(j)} = \mathbf{E}_j(z_j)$, then $\tilde{\mathbf{e}}_i = \sum_{j=1}^{n} \mathbf{e}_{z_j}^{(j)}$. Cross-entropy loss using $n$-gram token prediction task on $\{\tilde{\mathbf{e}}_i\}$ can be written as:

$$
\begin{aligned}
\mathcal{L}(\hat{\mathbf{h}}, x^{(+n)}; \tilde{\mathbf{E}}) &= -\log \frac{\exp(\hat{\mathbf{h}}\tilde{\mathbf{e}}_{x^{(+n)}})}{\sum_i \exp(\hat{\mathbf{h}}\tilde{\mathbf{e}}_i)} = -\log \frac{\exp(\hat{\mathbf{h}}(\sum_j \mathbf{e}_{x_j}^{(j)}))}{\sum_{z_1, \ldots, z_n} \exp(\hat{\mathbf{h}}(\sum_j \mathbf{e}_{z_j}^{(j)}))} \\
&= -\log \frac{\prod_j \exp(\hat{\mathbf{h}}\mathbf{e}_{x_j}^{(j)})}{\sum_{z_1, \ldots, z_n} \prod_j \exp(\hat{\mathbf{h}}\mathbf{e}_{z_j}^{(j)})} = -\log \frac{\prod_j \exp(\hat{\mathbf{h}}\mathbf{e}_{x_j}^{(j)})}{\prod_j \left(\sum_{z_j} \exp(\hat{\mathbf{h}}\mathbf{e}_{z_j}^{(j)})\right)} \\
&= -\log \prod_j \frac{\exp(\hat{\mathbf{h}}\mathbf{e}_{x_j}^{(j)})}{\sum_{z_j} \exp(\hat{\mathbf{h}}\mathbf{e}_{z_j}^{(j)})} = \sum_j -\log \frac{\exp(\hat{\mathbf{h}}\mathbf{e}_{x_j}^{(j)})}{\sum_{z_j} \exp(\hat{\mathbf{h}}\mathbf{e}_{z_j}^{(j)})} = \sum_j \mathcal{L}(\hat{\mathbf{h}}, x_j; \mathbf{E}_j),
\end{aligned} \tag{9}
$$

where $x_j$ is the next-$j$ token, and $x^{(+n)} = \sum_j x_j V^{j-1}$. Hence, with product embedding decomposition, the $n$-gram token prediction can be decomposed to multiple next-$j$ token prediction loss, which can be calculated efficiently.

**Discussion.** Based on product embedding decomposition, we propose a training objective similar to MTP (Gloeckle et al., 2024). Formally, we remove the additional transformer layer introduced by the multiple prediction heads in MTP. Instead, we directly decode the next 1 to $n$ tokens from the last hidden state using $n$ different vocabularies.

From the perspective of MTP, the nonlinear transformation introduced by directly using new vocabularies in the prediction heads is not weaker than adding a transformer layer. Moreover, since the unembedding process is a computational bottleneck, using a new set of parameters does not reduce training throughput compared to MTP.

From the perspective of Over-decoding, we retain the potential to scale up the output vocabularies. This means that the training process provides finer-grained supervision signals, which may enable the model to learn better representations. However, this approach is likely only practical for sufficiently large models. On one hand, larger models have a smaller proportion of computation spent on unembedding. On the other hand, larger models offer greater capacity to take advantage of this setup.

**Algorithm 2** Over-Decoding in a PyTorch-like style.

```
#OD parameters:
# n: the number of next tokens involved
# lambdas: list of n-1 elements, indicating loss weights for next i+1 token pred
# d: dimension for low-rank decomposition
#Model parameters:
# V: base vocabulary size
# D: model dimension

#Torch Modules
wte = nn.Embedding(V, d) # default next 1 token embedding table
od_embs = nn.ModuleList([
    nn.Sequential(
        nn.Linear(D, d) if D != d else nn.Identity(),
        nn.Linear(d, V)
    )
    for i in range(n-1)])
ce_loss = nn.CrossEntropyLoss()

def forward(self, input_ids, hidden_states):
    # input_ids: [bs, seqlens]
    # hidden_states: [bs, seqlens, D]
    loss = ce_loss(F.linear(hidden_states[:,:-1], wte.weight.T), input_ids[:,1:])
    for i in range(2, n+1):
        loss += lambdas[i-2] * ce_loss(od_embs[i-2](hidden_states[:-i]), input_ids[:,i:])
    return loss
```

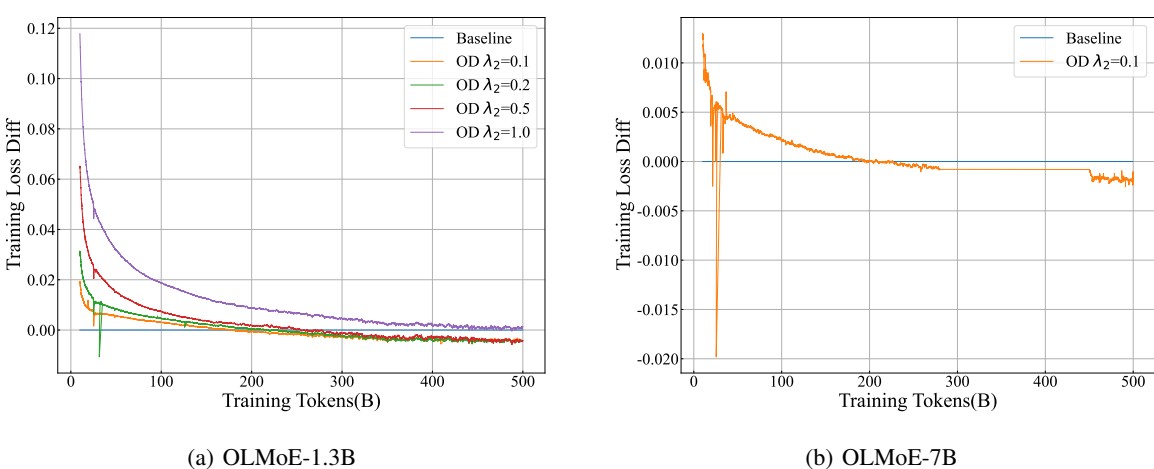

(a) OLMoE-1.3B

(b) OLMoE-7B

Figure 14: Loss diff curves comparing over-decoded models and baselines. OD models use the setting of $n = 2$ and $\lambda_2 \in [0.1, 0.2, 0.5, 1.0]$. The curves are smoothed with exponential moving average of weight $0.99$.

### C.1. Experiments

We conduct experiments for Over-Decoding under OLMoE settings, where both OLMoE-1.3B and OLMoE-7B are considered. We train 500B tokens for all the experiments.

**OD improves baseline with sufficient training.** We first observed that over-decoding significantly lags behind the baseline in the early stages of training. However, after training on more than 200B tokens, over-decoding begins to outperform the baseline. This experiment is conducted on both 1.3B and 7B OLMoE models, where we set $n = 2$. As shown in Figure 14, the next-one-token loss for over-decoding surpasses the baseline with a steeper slope. Downstream metrics, as illustrated in Table 10, also show slight improvements with proper loss weight choosen. Typically, we find training loss converges to similar values regardless of loss weights. But from the downstream tasks, a smaller weight leads to better overall performance.

Table 10: Ablation study on loss weights for OD. The column downstream represents the average score of MMLU-Var, Hellaswag, ARC-Challenge, ARC-Easy and PIQA.

| Model | Loss↓ | Eval Loss↓ | MMLU-Var↑ | Hellaswag↑ | Arc-Challenge↑ | Arc-Easy↑ | PIQA↑ |
|---|---|---|---|---|---|---|---|
| OLMoE-1.3B | 2.554 | 2.924 | 0.327 | **0.553** | 0.325 | 0.622 | 0.727 |
| +OD $\lambda_2 = 0.1$ | 2.549 | 2.920 | 0.325 | **0.553** | **0.331** | 0.610 | 0.721 |
| +OD $\lambda_2 = 0.2$ | 2.549 | **2.918** | 0.327 | 0.551 | 0.323 | **0.633** | **0.728** |
| +OD $\lambda_2 = 0.5$ | 2.549 | **2.918** | 0.325 | **0.553** | 0.308 | 0.619 | 0.727 |
| +OD $\lambda_2 = 1.0$ | 2.555 | 2.923 | **0.328** | 0.550 | 0.320 | 0.629 | 0.722 |
| OLMoE-7B | 2.306 | **2.670** | 0.385 | **0.695** | **0.414** | **0.740** | 0.775 |
| +OD $\lambda_2 = 0.1$ | **2.304** | 2.672 | **0.387** | 0.691 | 0.409 | 0.724 | **0.776** |

