# OpenReview forum: "Over-Tokenized Transformer: Vocabulary is Generally Worth Scaling"
_ICML.cc/2025/Conference — ICML 2025 poster_

### Official Review · Reviewer_wJC2 · 2025-03-11

**Overall Recommendation:** 3

**Summary:**

This paper proposed a new framework called over-tokenized transformer for language modeling, which decouples input and output vocabularies for performance improvement while leveraging the $n$-gram tokens. They have experimentally shown a log-linear relationship between the input vocabulary size and the model training loss and found that larger input vocabulary contributes to performance improvement, albeit larger output vocabulary would require larger model size due to over-fitting.
With extensive experimental results, this work highlights the importance of tokenization design in large language models, discussing the design of both input embedding and output unembedding.

**Claims And Evidence:**

With clear correlation between vocabulary size and training loss, their claim is well supported. Leveraging multi-gram tokens in language model is a straight forward idea, and this enhances better contextualization, resulting in lower training loss. On the other hand, I am very curious how this approach impact in terms of inference speed as the proposed approach significantly increase the model parameter size. Inference speed is pragmatically important, thus it would be interesting to compare each model in terms of the inference speed. Additionally, I feel this approach probably loses diversity in the search space at inference. If so, do you have any idea on how to diversify text generation?

**Essential References Not Discussed:**

- Liu et al. "Infini-gram: Scaling Unbounded n-gram Language Models to a Trillion Tokens" in Proc of COLM 2024.

**Experimental Designs Or Analyses:**

Please see the comments in Claims And Evidence and Methods And Evaluation Criteria.

**Methods And Evaluation Criteria:**

Does Figure 4 report all training losses/metrics for each task? How about the valid loss/metrics?

**Other Comments Or Suggestions:**

- l.104 (Tao et al., 2024) demonstrates that -> Tao et al.(2024) demonstrate that
- l.083 2.3. Multi-Token Prediction and n-Gram Modeling -> 2.3. Multi-Token Prediction and $n$-Gram Modeling
- Figure 4 - consider displaying the number (e.g., "5.7x") in a different position. Hard to read them.

**Other Strengths And Weaknesses:**

N/A

**Questions For Authors:**

Please see the questions in each section.

**Relation To Broader Scientific Literature:**

The idea of over-tokenization Transformer could be beneficial as the proposed approach is applicable to any types of large language models. It would be interesting to apply it into a multilingual language modeling.

**Theoretical Claims:**

N/A

---

> ### Author Rebuttal · Authors · 2025-03-30
>
> We appreciate the time and effort you have taken to review our manuscript, and are truly thankful to the reviewers for the insightful comments. As a response, we address each point individually.
> ## Analysis of the training & inference speed
> To illustrate training efficiency, we show training throughputs in OLMoE experiments in the following table, where we run OE and baseline under the same hardware configurations. OLMoE-7B yields more overhead as we did not carefully optimize engineering configurations.
>
> **Table**. Training throughputs for OE and baseline. We report average tokens per second in millions.
> |               | OLMоE-1.3B        | OLMоE-7B         |
> | ------------- | ----------------- | ---------------- |
> | Hardware      | 32  A100         | 64  A100        |
> | baseline  | 1.211           | 0.494          |
> | +OE 12.8M   | 1.155 (-4.63%)  | 0.453 (-8.3%)  |
>
> Theoretically, the additional flops introduced by OE is less than 0.5% (as shown in the table below). The overhead measured in the above experiments should mainly be introduced by the all-to-all communication (which is proportional to the size of data parallel). And these communication overheads can be further optimized through engineering techniques in the future.
>
> **Table**. FLOPs per token in the forward pass.
> |  | OLMoE-1.3B | OLMoE-7B |
> | --- | --- | --- |
> | baseline | 0.5409 G | 2.3578 G  |
> | +OE 12.8M | 0.5430 G (+0.38%) | 2.3662 G (+0.35%) |
>
> For inference speed, we tested the prefill and decoding throughput on a single A100 GPU using the `transformers` library. **For the OE models, the additional embedding parameters are offloaded to the CPU, incurring no GPU memory overhead**. The numeric results are shown in the following table. The impact of OE on inference throughput is negligible, especially for larger models or larger batch sizes. In contrast, the sparse parameters introduced by MoE face severe memory access bottlenecks during inference. A very large batch size is required for the MoE model to achieve the same throughput as a dense model with the same activated parameters. Considering that the model inference might be carried out on more cost-effective but less computationally powerful inference GPUs, the relative overhead of OE could be further reduced.
>
> **Table**. Inference speed for OE and baseline. The sequence length is fixed to 2048, and we report tokens per second for prefilling and decoding separately under various batchsizes. Settings that cause OOM are leave blank.
> |  |  | Dense-1B | Dense-1B | MoE-1B/7B | MoE-1B/7B | Dense-7B | Dense-7B |
> | --- | --- | --- | --- | --- | --- | --- | --- |
> |  |  | baseline | OE-12.8M | baseline | OE-12.8M | baseline | OE-12.8M |
> | bs=1| prefill | 20728.7 | 19446.4 | 6303.8 | 6189.0 | 6571.0 | 6499.9 |
> |  | decode | 136.2 | 126.6 | 28.2 | 27.9 | 65.1 | 63.3 |
> | bs=8 | prefill | 36907.5 | 35902.6 | 22297.3 | 22292.1 | - | - |
> |  | decode | 797.2 | 760.9 | 184.9 | 181.0 | 232.1 | 228.6 |
> | bs=64| prefill | - | - | - | - | - | - |
> |  | decode | 1422.1 | 1407.4 | 860.3 | 826.7 | - | - |
>
> These analysis will be added to our paper in camera-ready version.
>
> ## About Inference Diversity
> This is an interesting perspective. We believe that theoretically there should be no difference, and we haven't observed such a phenomenon in practice either. In fact, examples of synthetic data can well answer this question. In the CFG task, if the problem of decreased output diversity exists, the predicted next token probability should have a larger difference from the groundtruth distribution that is calculated according to the grammatical rules. We actually calculated the KL divergence in our experiments, and the OE also shows better performance compared to the baseline, indicating that OE can model the grammar better and generate sequences as diverse as the language itself can be.
>
> ## About Questions on Validation
> Regarding the experimental setup, we mainly followed the experiments of OLMo and used its training data and evaluation protocols. Specifically, the evaluations include: the validation sets of public text datasets (such as C4-en-validation), on which we calculate the next token prediction loss/perplexity as the evaluation metrics; and open benchmarks (such as hellaswag), on which we calculate the zero-shot accuracy.
>
> Figure 4 in the paper shows some of the evaluation set metrics that we are mainly concerned (explained in line 220~230). You can find comprehensive evaluation results on Figure 8 in the appendix, where we show the eval losses could has consistent gains.
>
> ## Writing Issues
> We appreciate the suggestions on paper writing. We'll improve the paper in the camera-ready version.

---

### Official Review · Reviewer_DdYQ · 2025-03-13

**Overall Recommendation:** 3

**Summary:**

This paper proposed methods to create much larger input/output vocabularies for transformers. For the input, (causal) n-grams embeddings are used. These are hierarchical in that they are the sum of n-grams for multiple values of n, including the original single valued token. Similarly they suggest Over Decoding where n-grams are predicted (but only the initial token is used as the auto-regressive input). They find that while Over Encoding helps in most settings, Over Decoding is only helpful for large models.

They evaluate Over Encoding based on c4 language modeling an a handful of downstream tasks and find that it yields significant gains in performance and training speed.

They also do ablations about which parts of their Over Encoding scheme are the most important and find that the hierarchical nature is important and that hash collisions should be minimized.

**Claims And Evidence:**

There claims are supported, their method shows strong improvement in multiple setting with multiple models and seems like they would extend to other seeings.

However, some claims in the probe like "insight for tokenizer design" and mentions of "more efficient LLMs" seem like overreach given they don't actually reduce sequence lengths or change how tokens created, instead their method is more akin to a specialized input layer that explicitly models things like n-grams.

**Essential References Not Discussed:**

Their "General n-gram Embedder" which looks up indexes modulo m such that different token could share an index is very similar to the hash-based embeddings from works like vowpal wabbit and fasttext. The study of how this embedding technique interacts with transformers in novel, but these works should be discussed with respect to the method.

The dense projections in their Over-Encoding section is very similar to the soft prompt reparameterization in works like https://arxiv.org/pdf/2101.00190 and https://arxiv.org/abs/2305.03937 so they should be mentioned.

**Experimental Designs Or Analyses:**

In figure 4 there are claims about "convergence acceleration", however, from the training loss it seems like their models have not converged, i.e., the training loss is still decreasing. These speed ups should probably be framed as time to reach the performance as the baseline models instead.

It would be nice to see experiments that teased apart if the gains are from having such a larger vocabulary or from the explicit modeling of n-gram composition. It would have been nice to see an ngram representation made of the sum of each token in the n-gram with shared embeddings instead of making new unrelated embeddings for "the" and "the fox" (only the "fox" token is included in the hierarchical embedding).

**Methods And Evaluation Criteria:**

Yes, they include both intrinsic evaluation on c4 language modeling and extrinsic evaluation on other datasets.

**Other Comments Or Suggestions:**

To me, it seems that the title of the paper "Vocabulary is Generally Worth Scaling" implies something different that what is studied. The title suggests a study that scale the size of the vocabulary meaning more unique words/types would be in the vocab (e.g. more merge operations are done in BPE so that less frequent words are still getting their own tokens). Their approach is more akin to making bigram and trigram tokens which was unexpected.

Indexing the ablation studies as things like C-3 makes reading difficult as it tells you nothing about what is being ablated.

On line 096 they refer to this work as "n-gram patching" similar to BLT. However, the "patching" in BLT is about reducing the sequence length while this work maintains the original sequence length, but has causal n-gram embeddings.

**Other Strengths And Weaknesses:**

The left side of figure 3 is very similar to standard transformer data flow diagrams (like the one on the right) but is communicating how the tokens are used. This makes it very confusing as it is read very differently from the right side. Also having Input tokens on the top of the right hand part of the image is non-standard which makes it a bit harder to read than a standard input on bottom image.

**Questions For Authors:**

Did you explore anything like what kind of text benefits the most from over encoding? For example, on average do you see a large boost in performance when processing a word like "unfortunate" that was broken into multiple tokens, i.e. ["un", "fortunate"] where the n-gram token is simulating what happens if that work wasn't split into subwords?

**Relation To Broader Scientific Literature:**

Recent works like https://arxiv.org/abs/2405.05417 discuss how under-trained tokens in LLMs can be used to facilitate unwanted model behavior. Using Over-Encoding means there will most likely be far more under-trained tokens (not only unseen unigrams but unseen bi or tri grams). It seems so analysis on how a technique like this effects LLM safety would be prudent.

**Theoretical Claims:**

N/A

---

> ### Author Rebuttal · Authors · 2025-03-30
>
> We appreciate the time and effort you have taken to review our manuscript, and are truly thankful to the reviewers for the insightful comments. As a response, we address each point individually.
>
> ## About Questions on Claims And Evidence
> As far as the over-encoding technique itself, indeed, it is typically a network module that improves performance and is irrelevant to the tokenizer. However, we believe the insight behind is worth noticing for the tokenizer design:
> 1. We show that the encoding and decoding in tokenization should be separately considered to maximize model performance. One should be careful in increasing decoding vocabulary.
> 2. Hierarchical designs improve OE, but can multi-granularity tokens be properly considered in tokenizer design itself?
> By introducing OE, we show that simply leveraging hashed n-gram tokens could yield significant improvements to LLM. We could expect this gain to be pushed further with delicate tokenizer designs.
>
> As for the "more efficient LLM", we mainly consider our efficiency on the proper use of sparse embedding parameters, similar to MoE, which improves model performance with negligible training and inference overheads. It is efficient as you can obtain powerful models with possibly half training and inference budget.
>
> ## About tease apart the gains.
> The reviewer mentioned:
> > It would be nice to see experiments that teased apart if the gains are from having such a larger vocabulary or from the explicit modeling of n-gram composition. It would have been nice to see an ngram representation made of the sum of each token in the n-gram with shared embeddings
>
> Actually, we tried such an approach in the early stage of our research. Summing n-gram tokens from a shared embedding doesn't work because there is no positional distinction for each token in the n-gram (addition is commutative). This makes the current input token unclear, which is detrimental to the model's performance. If we simply remove the condition of sharing embeddings and apply n different embedding tables to these n tokens, the model can achieve some benefits. Later, we found that the gains from this implementation are close to those predicted by the OE scaling curve with \(m=(n - 1)\times V\). Essentially, this implementation is a product decomposition for the full n-gram embedding table. Alternatively, you can also view it as OE using a special hash function, and the choice of hash function isn't crucial for OE.
>
> In addition, our ablation study has also ablated the effect of increasing vocabulary size and n-gram composition. In Figure 5, we fix n=2 and varied vocabulary size, showing the gains from scaling vocabulary size. In Table 3, we fix the embedding parameters and ablated gains introduced by hierarchical n-gram composition. We conclude that both vocabulary size and well-designed n-gram composition are important to OE.
>
> ## About the under-trained tokens
> It's an interesting perspective to consider under-trained tokens. However, we believe OE should not have such a problem. Under-trained tokens are potentially harmful mainly due to their under-trained embedding vectors. For OE, n-gram embeddings are equally visited during training owing to the many to one hashing. Though there might be some unknown n-gram tokens occur during inference, their embedding vectors are guaranteed to be frequently trained. So, we believe the unseen n-gram tokens are more like a normal spelling mistake and will not cause catastrophic consequences. Moreover, the token embeddings should contain at least one well-trained uni-gram embedding under the hierarchical design, which also improves robustness against unseen n-grams.
>
> ## What kind of text benefits the most from over-encoding
> Examining how n-gram embedding contributes to some specific token is kind of difficult. As the n-gram embeddings improve the keys and values in the attention layer simultaneously. As a result, it is difficult to tell where the improvements on specific tokens come from. However, from the evaluation results, we do notice that OE has significantly larger improvement in knowledge-related tasks (see few-shot results in our response to Reviewer ye13). We hypothesis that coarse-grained token embeddings help memorizing concepts of proper nouns, which are usually broken into several tokens under the base tokenizer.
>
> ## About the Relevance of BLT
> We are sorry for the word 'patching' that misleads the understanding. We typically want to talk about the n-gram embedding technique used in BLT. They apply n-gram byte embeddings to the byte-level sequence, which does not reduce byte sequence length as well. We'll revise the paper to make this more clear.
>
> ## Related Work & Writing Issues
> We appreciate the suggestions on related work and paper writing. We'll improve the paper in the camera-ready version.

---

### Official Review · Reviewer_ye13 · 2025-03-14

**Overall Recommendation:** 5

**Summary:**

This paper introduces a novel method of scaling vocabulary size for LLMs, where given an existing tokenizer, the model constructs n-gram representations on-the-fly. Several algorithmic optimizations (matrix decompositions) are made to limit the size of the embedding table while handing the exponential growth of n-gram embeddings. Namely: n-gram representations are tiled across a fixed vocabulary size, m, and their hidden dimensions are low-rank -- only projected back to d_model when needed. m then becomes a knob to control how much memory one wants to allocate to n-gram representations. This decomposition though preserves the sparse lookup nature of embedding table, preventing massive FLOP overhead (although there is some overhead from the new projection). Extra memory costs are handled by hardware/engineering optimizations to shard the memory across many GPUs. Authors show that scaling m, and introducing n-gram representations greatly improve performance on both synthetic, training dynamics, and 0-shot downstream tasks, for OLMo2 models up to 1B dense and 7B MoE (~1B activated). Authors show that there exists a log-linear relationship between scaling vocabulary size and training loss. Vocabulary size is able to be scaled up to 12.8M.

**Claims And Evidence:**

Claims in this paper are generally sound and the authors provide compelling empirical evidence of the effectiveness of their method. They show that the method works across a wide array of experimental settings at reasonably large scale: synthetic, training loss, holdout perplexity, 0-shot downstream, for models up to 1B Dense and ~1B activated 7B sparse MoE, for 500B tokens. To the best of my understanding, the authors do not overclaim any aspect of their paper. The results genuinely look quite strong. Authors report results even when not the most flattering for their method (OLMoE-7B, Table 1).

**Essential References Not Discussed:**

Related work is a bit sparse.

Prior works that have aggregated character-level n-grams in different ways are missing:
- CANINE https://arxiv.org/abs/2103.06874
- Charformer https://arxiv.org/abs/2106.12672
Authors only cite MegaByte, which is not the first instance of this. Early works in character-models discussed above also previously pioneered decoupling of input and output vocabularies.

Authors are missing some of the very first examples of MTP, such as blockwise decoding (2018): https://arxiv.org/abs/1811.03115

**Experimental Designs Or Analyses:**

Yes, the experimental design is sound: all ablations are done on completely fixed settings, ablating only the proposed change to the vocabulary and comparing against the baseline. I do not see any issues with the author's analysis, see "Claims And Evidence" section.

**Methods And Evaluation Criteria:**

Yes, the authors relatively large scale models from scratch, using standard OLMo settings comparable to the baseline. For evaluation, the use of holdout perplexity is meaningful for architecture design. Moreover, the authors provide a full suite of zero-shot tasks, which is sufficient for this work. It would strengthen the work also to include few-shot tasks, to demonstrate that the method preserves in-context learning abilities.

A major motivation of this paper is to show the value of scaling vocabulary size / parameters. The authors also effectively show this in log-linear relationship in Figure 5, which is quite compelling.

**Other Comments Or Suggestions:**

Description and notation of the method could be written to be easier to follow. In particular the use of excessive superscripts, and E -- typically reserved for expected value, is a little confusing. The i in the summation of Eq (5) isn't passed anywhere.

**Other Strengths And Weaknesses:**

Strengths of this paper has been well covered in previous sections. This is an exciting, novel, modeling improvement that has the capability of unlocking new scaling avenues in the field. The empirical results are thorough and convincing.

Weaknesses: More thorough analysis of the training and inference speed would be welcome, as this could be a major blocker for adoption. This would strengthen the efficiency results reported in Section 3.3. Especially as the OE method is light on additional FLOPs, but not FLOP-neutral (extra up-projection).

**Questions For Authors:**

Is there a slowdown in walltime for training, or the ms/step for decoding?

Both OE and MoE incurs memory cost, is there any advantage in preferring one type of sparsity for another? Do they unlock different types of capabilities? How would one choose the sparsity tradeoff given fixed memory budget if combining both MoE and OE?

**Relation To Broader Scientific Literature:**

While as the authors note that there has been studies regarding scaling vocabulary size in the past. To the best of my knowledge, this work is quite novel for a variety of reasons:

1. It uses an unigram vocabulary as the baseline, as it does not change the tokenizer.
2. It dynamically allows for control over the extra sparse capacity allocated for n-gram representations, similar to sparse methods like MoEs.
3. It's the first time that vocabulary can be dynamically grown in the model, and the quality is shown to improve with larger dynamic vocabularies.

The closest work to this in the byte-level transformer space, which tries to do similar at the byte level (aggregate bytes into byte-grams or tokens), however this is typically with the goal of improving byte-level models to match token-level performance. Here, we show that token-level models gain even more performance by going n-gram.

This an exciting new direction in the field and could open up new avenues for modeling scaling, worthwhile of further exploration.

**Theoretical Claims:**

The authors do not make any theoretical claims. The description of their method is sound.

---

> ### Author Rebuttal · Authors · 2025-03-30
>
> We appreciate the time and effort you have taken to review our manuscript, and are truly thankful to the reviewers for the insightful comments. As a response, we address each point individually.
>
> ### Performance on Few-shot Tasks
> We have conducted few-shot evaluations for in-house experiments. Our in-house baseline follows MoE architecture with 400M activated parameters and a total of 4B sparse parameters. And we implement OE to scale up embedding table m to 36M. Here, we share some of the results.
> #### Reasoning Related Benchmarks:
> |                | ARC Challenge | Drop  | BBH   | WinoGrande | Hellaswag |
> | -------------- | ------------- | ----- | ----- | ---------- | ------- |
> | Baseline | 65.7          | 34.4  | 37.1  | 63.2       | 66.2    |
> | +OE 36M        | 67.9          | 36.3  | 39.5  | 65.5       | 67.2    |
>
> #### Knowledge Related Benchmarks:
> |                | MMLU  | C-Eval | TriviaQA | MMLU-Pro | AGIEval |
> | -------------- | ----- | ------ | -------- | -------- | ------- |
> | Baseline | 54.8  | 61.3   | 39.7     | 21.1     | 39.1    |
> | +OE 36M        | 57.9  | 68.3   | 49.0     | 24.1     | 43.2    |
>
> #### Math Related Benchmarks
> |                | Ape210K | GSM8K | MATH |
> | -------------- | ------- | ------ | ---- |
> | Baseline | 63.7    | 40.6   | 22.7 |
> | +OE 36M        | 63.8    | 46.2   | 25.3 |
>
> We will add these results to our paper in the camera-ready version.
>
>
> ### Analysis of the training & inference speed
> We put thorough analysis on response to Reviewer wJC2. In conclusion, OE has 4% and 8% training overhead on OLMoE 1.3B and OLMoE 7B. We emphasize the larger overhead in OLMoE7B is owing to that we did not carefully optimize engineering configurations. Theoretically, OE only introduces less than 0.4% FLOPs. The overhead measured in the experiments should mainly be introduced by the all-to-all communication (which is proportional to the size of data parallel). And these communication overheads can be further optimized through engineering techniques in the future.
>
> As for inference, we tested the prefill and decoding throughput on a single A100 GPU using the transformers library. For the OE models, the additional embedding parameters are offloaded to the CPU, incurring no GPU memory overhead. The impact of OE on inference throughput is negligible (around 2%).
>
>
> ### How would one choose the sparsity tradeoff given fixed memory budget if combining both MoE and OE?
> It's an interesting question. First of all, it must be admitted that the performance of OE with the same number of sparse parameters is not as good as that of MoE. However, we'd like to emphasize that scaling up the parameters of MoE is not free. During the inference, MoE increases the GPU memory overhead. Moreover, under small batch sizes, there are memory access bottlenecks, and the inference efficiency is usually far from reaching that of a dense model with the same activated parameters (as shown in the inference speed table on our response to Reviewer wJC2). In contrast, during the inference process, OE can be completely offloaded to the CPU, incurring no GPU memory overhead, and the reduction in efficiency is almost negligible. Our recommended approach is to first determine the size of MoE according to the inference requirements, and then use the remaining GPU memory to scale the embedding table.
>
> In addition, considering that OE has a good property during the training phase: the input of OE only depends on the token id, it provides more room for engineering optimization. For example, during the training phase, OE can possibly be offloaded to the CPU and the embeddings of the next micro batch can be prefetched and overlap with current micro batch's forward. On this occasion, OE and MoE can operate independently without interfering with each other. We are also exploring different engineering solutions for this issue.
>
>
> ### Related work & Equation Typos
> Thanks for your careful reading, equation 5 has some typos. The correct formula should be:
> $$\texttt{OE}(x)= \mathbb{E}^{V\times d}(x^{(-1)}) +\sum_{i=2}^{n} \mathbb{E}^{m\times \frac{d}{n}|k}(x^{(-i)})$$
> And we appreciate the suggestions on paper writing and related work. We will improve the paper in the camera-ready version.

---

### Official Review · Reviewer_rXP1 · 2025-03-14

**Overall Recommendation:** 4

**Summary:**

This paper reveals the scaling law of vocabulary size. They decouple the encoding and decoding vocabulary and introduce Over-Tokenized Transformers. Using CFG, they demonstrate the advantages of larger vocabulary size in synthetic settings. With this intuition, they design and train language models with larger encoding vocabulary. They show that larger vocabulary leads to clear improvement on model performance with less training steps.

**Claims And Evidence:**

1. Under the CFG task, the larger models can benefit from larger vocabulary size, but small models do not. Moreover, the scaling up of encoder vocabulary size is better than the scaling up of decoder size. Although Figure 2 perfectly demonstrates this claim, I feel that it would be better to provide more fine-grained analysis on these claims. The synthetic nature of the CFG task may make it possible for more fine-grained analysis.

2. The authors further show that scaling up the encoding vocabulary size with n-gram for language models can give much faster convergence for the training loss. It also improves the performance on models on benchmarks. They adopt a variety of metrics and implement many ablation studies. I find these claims to be convincing.

**Essential References Not Discussed:**

Not sure.

**Experimental Designs Or Analyses:**

No.

**Methods And Evaluation Criteria:**

The combination of loss/perplexity and the scores on benchmarks make the results convincing.

**Other Comments Or Suggestions:**

There seems to be some typos. Equations (4) and (5) are confusing. The texts seem to indicate that all k-gram encodings will be used (k=1,..,n). But the equations conflict with the description.

**Other Strengths And Weaknesses:**

1. The tokenization plays a key role in language modeling, but is not well-discussed in the literature. The authors provide some valuable insights for better understanding of this crucial component of LLMs.
2. Can authors comment on the choice of using n-gram for scaling-up vocabulary size? A natural way of scaling up vocabulary size would be keep running the BPE algorithm and get more tokens. This approach also makes more sense since it is adaptive to the data distribution. Directly merging tokens to be n-grams will lead to many "wasted" tokens.
3. Although the scaling law seems to be clear, the slope of the scaling line is too small, making the overall improvement seems to be negligible. The loss and the scores of the OE models are still very close to the base model.

**Questions For Authors:**

No

**Relation To Broader Scientific Literature:**

Not sure.

**Theoretical Claims:**

N/A

---

> ### Author Rebuttal · Authors · 2025-03-30
>
> We appreciate the time and effort you have taken to review our manuscript, and are truly thankful to the reviewers for the insightful comments. As a response, we address each point individually.
>
> ### The choice of using n-gram tokens.
> Continuing to train BPE to obtain a larger vocabulary is indeed a more natural approach, and we have also attempted such a practice in our early experiments, where we keep frequent bigrams to construct a large one-to-one embedding table and discard unfrequent bigrams. However, the experimental results show that this approach is not better than simply hashing. This may be because the frequencies of the expanded embeddings differ too greatly (the hottest and the coldest can differ by a factor of 10,000), resulting in the parameters of the expanded embeddings not being sufficiently trained. In contrast, the hashing-based method can ensure that all expanded embeddings have an equal access frequency, thereby ensuring sufficient training and better scalability. Of course, we also encourage future work to continue exploring in this direction and to achieve end-to-end encoding of multi-granularity tokens directly from the tokenizer.
>
> ### The slope of the scaling line is too small.
> As for the slope of the embedding parameters' scaling curve, we are not proposing to replace the scaling law of the standard dense parameters, but rather to provide an additional, nearly **COST-FREE**, second growth curve. From this perspective, as long as this scaling benefit exists, it remains a better solution.
> Typically, you can keep scaling up the embedding parameters with negligible inference cost and barely any additional training costs other than the GPU memory usage (please refer to our response to Reviewer wJC2 for numberic results). To inference, the large embedding table can simply be offloaded to CPU. And in fact, the issue of GPU memory overhead during training can also potentially be addressed through CPU offload and prefetch (might be solved in future work). Under these circumstances, as long as we scale it up to a large enough extent, e.g., 128 times the original vocabulary size, we can achieve significant performance improvements, e.g., 400M OE loss is on par with the 1B baseline.
>
> ### Typos in Equations.
> Thanks for your careful reading. We apologize for the confusion on our formulations. Yes, there is a typo in equation (5). We leverage both 1-, 2-,.., n-gram tokens, so the equation is expected to be
> $$\texttt{OE}(x)= \mathbb{E}^{V\times d}(x^{(-1)}) +\sum_{i=2}^{n} \mathbb{E}^{m\times \frac{d}{n}|k}(x^{(-i)})$$
> As for parameter k, it indicates a slicing factor. And flattening this equation should result in:
> $$\texttt{OE}(x)= \mathbb{E}^{V\times d}(x^{(-1)}) +\sum_{i=2}^{n} \sum_{j=1}^{k}\mathbb{E}^{m\times \frac{d}{nk}}(x^{(-i)})W_{i,j},$$ where $$W_{i,j}\in \mathbb{R}^{\frac{d}{nk}\times d}$$
> We hope this could make the formulation clear. We'll revise the paper in the camera-ready version.

---

### Decision · Program_Chairs · 2025-05-01

**Decision:**

Accept (poster)

**Comment:**

This paper introduces a method for decoupling input and output vocabularies. The authors propose to scale input vocabularies using hierarchical, hashed n-grams, showing strong empirical results on large models, and significant performance improvement on smaller models. The results show a log-linear relationship between input vocabulary size and training loss. Reviewers all agreed on the novelty, strong results, and potential impact of this work. Initial concerns regarding clarity (notation, figures), wall-time, and missing related work were mostly addressed in the authors' rebuttal. Despite other minor suggestions from the reviewers, the core contribution is sound and significant for ICML.